# Dinuclear gold-catalyzed divergent dechlorinative radical borylation of *gem*-dichloroalkanes

Cheng-Long Ji [1], Hongliang Chen[1], Qi Gao[1], Jie Han[1], Weipeng Li [1] & Jin Xie [1] ✉

The enormous and widespread use of organoboronic acids has prompted the development of innovative synthetic methodologies to meet the demands on structural diversity and functional group tolerance. The existing photo-induced defunctionalization radical borylation, typically focused on the conversion of one C-X bond (X= Br, I, or other leaving group) into only one C-B bond. Herein, we disclose a divergent radical dechloroborylation reaction enabled by dinuclear gold catalysis with visible light irradiation. A wide range of structurally diverse alkyl boronic, $\alpha$-chloroboronic, and *gem*-diboronic esters can be synthesized in moderate to good yields (up to 92%). Its synthetic robustness is further demonstrated on a preparative scale and applied to late-stage diversification of complex molecules. The process hinges on a C-Cl bond relay activation in readily available *gem*-dichloroalkanes through inner-sphere electron transfer, overcoming the redox potential limits of unreactive alkyl chlorides.

Boron, with an electronegativity of 2.0, has endued corresponding boronic acids and their esters to be one kind of useful functional groups[1]. These boron-containing compounds have found extensive applications in organic synthesis[2], material science[3], medicinal chemistry[4], analytic chemosensing[5], and chemical biology[6]. Notably, they are key coupling partners in Suzuki coupling reactions for carbon-carbon bond formation in organic synthesis and drug discovery[2]. As Lewis acid, boronic acids facilitate transitions between planar tricoordination and tetrahedral tetracoordination boron-center via reversible covalent bonding features[7], holding many advances in self-assembled organic materials, chemoprobes and pharmacodynamic activities[3,5,6] Moreover, the functional group of boronic acid can serve as ideal bioisostere for carboxylic acids, enhancing physiochemical properties in medicinal chemistry[4]. Consequently, facile installation of boronic acids from readily available chemicals is of significant synthetic importance, allowing for the widespread use of boronic acids in many fields of science (Fig. 1a). While radical borylation has advanced the production of aromatic boronic acids[8–10], the catalytic radical borylation of C(sp³)-rich materials is still in its infancy[11–13].

Alkyl halides, commonly used in organic chemistry, are ideal radical precursors for the preparation of alkyl boronic acids. Alkyl chlorides, more abundant and economical than alkyl iodides and bromides, have been less utilized due to their higher bond dissociation energy (BDE) (-84 kcal mol⁻¹) and lower reduction potential ($E_{red}$ = ~ −2.8 V versus SCE)(Fig. 1b)[14]. Traditional stoichiometric methods, involving reactive organometallic reagents (alkyl-Li or alkyl-MgX), have limited the functional group tolerance[15]. Given the extensive use of alkyl boronic acids, there is a growing need for synthetic methodologies that can accommodate structural diversity and functional group tolerance. Transition metal-catalyzed radical borylation has become an emerging protocol but generally requirement of strong organic base[16–22]. For example, the Marder's group disclosed several examples on Cu(II)-catalyzed *gem*-diborylation of dichloroalkanes in the presence of stoichiometric MeOK[17]. Recent advances in B-B bond activation include the Lu group's electrochemical *gem*-diborylation of highly active *gem*-bromoalkanes via paired electrolysis[23], as well as photo-induced defunctionalization radical borylation, which offers milder conditions and better versatility[24]. As illustrated in Fig. 1c, although

[1]State Key Laboratory of Coordination Chemistry, Jiangsu Key Laboratory of Advanced Organic Materials, Chemistry and Biomedicine Innovation Center (ChemBIC), School of Chemistry and Chemical Engineering, Nanjing University, Nanjing 210023, China. ✉e-mail: xie@nju.edu.cn

**a** Current status of construction and applications of boron-containing compounds

**b** Chemical features and challenges of alkyl halides

**c** The state-of-the-art for photoinduced defunctionalization radical borylation

**d** This work: Photoinduced gold-catalyzed divergent dechloroborylation of *gem*-dichloroalkanes

**Fig. 1 | Challenges and strategies in radical borylation of unactivated C-Cl bonds. a** Current status of construction and applications of boron-containing compounds. **b** Chemical features and challenges of alkyl halides. **c** The state-of-the-art for photoinduced defunctionalization radical borylation. **d** Photoinduced gold-catalyzed divergent dechloroborylation of *gem*-dichloroalkanes.

$C(sp^3)$-X (X = Br, I or other leaving groups) have been widely explored in visible-light-induced defunctionalization radical borylation[11,25–29], few photocatalytic strategies are capable of homolytic conversion of $C(sp^3)$-Cl bonds for radical borylation[30,31]. Inspired by the seminal works on gold-catalyzed C-X (X= Br, I) bond activation[32–39], we have demonstrated that using dinuclear gold complexes and ligand micro-environmental change strategy can circumvent the high reduction potential of alkyl chlorides through inner-sphere electron transfer mechanism[40]. As a result, the unreactive *gem*-dichloroalknes, which are usually prone to proceed hydrogen atom transfer (HAT) reactions[41], construct carbon-carbon bonds with various alkenes. In this context, we surmised the possibility of a divergent radical dechloroborylation protocol of readily available *gem*-dichloroalkanes, which will further expedite the next generation of borylation in the future (Fig. 1d). Its success can provide a direct and practical route to convert $C(sp^3)$-Cl into a wide range of structurally diverse alkyl boronic, $\alpha$-chloroboronic and *gem*-diboronic esters, and the scope is far beyond with previous defunctionalization radical borylation.

In this study, we herein disclose a dechlorinative radical borylation of *gem*-dichloroalkanes through photoexcited dinuclear gold catalysis. This method effectively utilizes a wide range of *gem*-dichloroalkanes, including dichloromethane, as starting materials. They can smoothly proceed gem-diborylation, monoborylation, and hydroborylation to produce structurally diverse high-value alkyl *gem*-diboronic, $\alpha$-chloroboronic and boronic esters in moderate to good yields under mild conditions. Interestingly, employing a continuous-flow technique further enhances the efficiency of the dechloroborylation reaction. This demonstrates its unique reactivity and excellent functional group compatibility, thus facilitating the late-stage borylation of complex molecules.

## Results

### Reaction development

The reaction development was initiated by using commercially available (3,3-dichloropropyl)benzene (**1a**) and 2,2'-bis-1,3,2-benzodioxaborole ($B_2cat_2$) (**2**) as the model substrate. After investigation of various photocatalysts, boron sources, and solvents, the optimal conditions include the use of dinuclear gold-complex (**PC1**) as photocatalyst and $B_2cat_2$ as the radical boron-source with *N,N*-diethylformamide (DEF) as the solvent at room temperature, which can deliver the desired product (**3**) in 73% isolated yield (Table 1, entry 1). Notably, this kind of *gem*-diboronic esters[42–47] are versatile building blocks in organic synthesis. When trinuclear gold-complex (**PC2**)[48] was employed, only a trace amount of desired products were detected (Table 1, entry 2). [Au(dppm)Cl]$_2$ (**PC3**)[49–51] was employed under the same conditions, and the yield significantly decreased to 45% (Table 1, entry 3). Other commonly used photocatalysts, such as *fac*-Ir(ppy)$_3$, Ru(bpy)$_3$Cl$_2$ and Eosin Y, all failed to achieve this transformation (Table 1, entries 4-6), possibly owing to their mechanistic limits to outer-sphere electron transfer[52]. Switching $B_2cat_2$ to $B_2pin_2$ led to no reaction (Table 1, entry 7). We speculated that the important role of the electron-rich aromatic diol ligand, catechol, on the boron reagent favors the radical borylation. Interestingly, it is found that DMF and NMP can also trigger radical borylation while MeCN gave rise to little products (Table 1, entries 8-10). We envisioned that the oxygen atom in DEF might coordinate with $B_2cat_2$[11], enabling the oxidation peak of $B_2cat_2$ shifted to more negative values, which can be rationalized by UV-Vis spectroscopy analysis, $^{11}$B NMR detection, and cyclic voltammetry (CV) experiments (see Supplementary Fig. 34, 46, and 56). In addition, it is found that the good solubility of DEF can give better results. The replacement of blue LEDs ($\lambda_{max}$ = 466 nm) with purple LEDs ($\lambda_{max}$ = 390 nm), a decreased yield was obtained (Table 1, entry 11). Control experiments showed that the absence of either the

## Table 1 | Optimization of the reaction conditions[a]

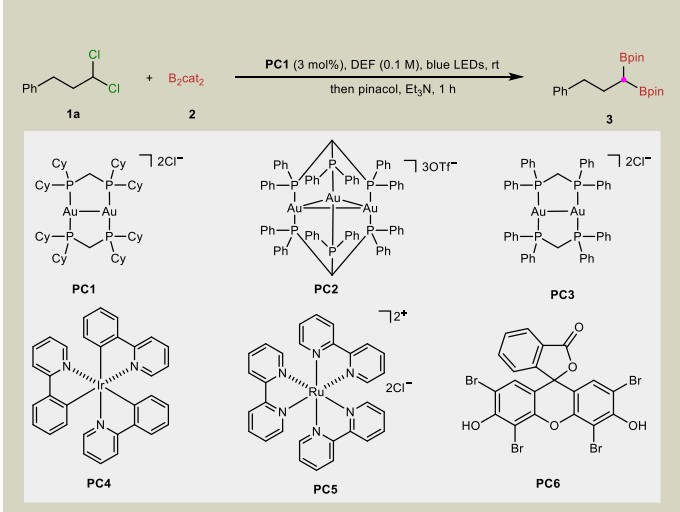

| entry | variation of standard conditions | yield (%)[b] |
|---|---|---|
| 1 | none | 78(73)[c] |
| 2 | **PC2** instead of **PC1** | trace |
| 3 | **PC3** instead of **PC1** | 45 |
| 4 | **PC4** instead of **PC1** | n.d. |
| 5 | **PC5** instead of **PC1** | n.d. |
| 6 | **PC6** instead of **PC1** | n.r. |
| 7 | B₂pin₂ instead of B₂cat₂ | n.r. |
| 8 | DMF instead of DEF | 47 |
| 9 | MeCN instead of DEF | n.r. |
| 10 | NMP instead of DEF | 27 |
| 11[d] | Purple LEDs | 52 |
| 12 | Without **PC1** | n.r. |
| 13 | Without light irradiation | n.r. |

[a]Standard reaction conditions: PC1 (3 mol%), **1a** (0.2 mmol), B₂cat₂ **2** (0.8 mmol), DEF (2 mL), blue LEDs ($\lambda_{max}$ = 466 nm), ambient temperature, 24 h; then pinacol (4.0 equiv), Et₃N (0.5 mL), 1 h. [b]GC yield using biphenyl as internal standard; [c]Isolated yield; [d]$\lambda_{max}$ = 390 nm. n.d. = not detected; n.r. = no reaction; DEF = N,N-diethylformamide; DMF = N,N-dimethylformamide; NMP = N-methyl-2-pyrrolidone.

photocatalyst or light resulted in no products, indicating the necessity of these components in this dechlorinative *gem*-diborylation reaction (Table 1, entries 12 and 13).

### Substrate scope

With the optimal conditions in hand, we next investigated the substrate scope and generality of this dechlorinative diborylation protocol (Fig. 2). A series of (3,3-dichloropropyl)benzene derivatives could deliver the corresponding *gem*-diborylalkanes (**3-13**) in moderate to excellent yields. Interestingly, aryl rings bearing with fluoro- or chloro-substituents (**8, 9**) were tolerated well, facilitating downstream modification at the halogenated position. In addition, aryl rings bearing electron-withdrawing or electron-donating groups at either *ortho*-, *meta*- or *para*-positions delivered the desired products (**6, 12, 13**) in 56-85% yields. Importantly, the protocol exhibited excellent compatibility with useful functional groups, including nitriles, esters, and furan (**10, 11, 14**). Notably, substituted *gem*-dichloroalkanes are tolerated in this transformation and furnished the corresponding borylation products (**15-17** and **20**) in moderate yields. A range of densely functionalized *gem*-dichloroalkanes containing unsaturated bonds, such as internal alkyne, terminal, or internal alkenes, can uniformly proceed gold-catalyzed diborylation to give rise to products (**18-21**) in 63-71% yields.

Intriguingly, the steric hindrance effect hardly influences this dechlorinative *gem*-borylation efficiency. For example, when the methyl group was introduced at the β-position of *gem*-dichloroalkanes, the desired products (**22-26**) were obtained in 48-75% yields. The other mono-substituted *gem*-dichloroalkanes can also proceed with this transformation smoothly to generate *gem*-diboronic esters (**27** and **28**) in moderate yields. Moreover, the highly sterically hindered double alkyl-substituted *gem*-dichloroalkanes, 4,4-dichlorotetrahydro-2H-pyran was subjected to the standard conditions, and it can deliver the expected product (**29**) in 70% yields. To demonstrate the practicability of this transformation, commercially available dichloromethane and deuterium-labelled dichloromethane also could be used to produce corresponding diboronic esters (**30** and **31**), which further illustrates that this strategy has promising advantages in organic synthesis.

Late-stage borylation of complex molecules represents the state-of-the-art in this field since the installation is generally at the early stage of synthesis owing to the technical difficulties in their preparation, thus severely compromising its total synthetic economy. In this protocol, gold-catalyzed dechlorinative *gem*-diborylation holds excellent functional group compatibility. A series of natural products and pharmaceuticals derivatives, including mitotan, oxaprozin, D-alpha-tocopherol succinate, linoleic acid, and chlorambucil were all suitable coupling partners and tolerated the conditions well, furnishing the corresponding products (**32–36**) in moderate yields.

With the success of gold-catalyzed dechlorinative *gem*-diborylation, we wonder whether the monoboronation of *gem*-dichloroalkanes could be achieved. If successful, it will provide a platform for the synthesis of substituted α-chloro alkylboronic esters, which are versatile building blocks in organic synthesis due to their amphiphilicity[53]. We found that gold-catalyzed dechlorinative monoborylation of very sterically hindered *gem*-dichloroalkanes can proceed smoothly, delivering the desired α-chloro alkylboronic esters in moderate to excellent yields (Fig. 3). A series of substituted cyclopropanes can successfully generate the corresponding monoborylation products (**37** and **38**). *Gem*-dichloroalkanes bearing fused rings were employed to react with B₂cat₂ (**2**) to afford the desired products (**39–41**) in 41–73% yields. The Bpin ester can be conveniently hydrolyzed to the corresponding boronic acid (**39a**). Furthermore, two alkyl groups were introduced at the β-position of *gem*-dichloroalkanes and furnished the monoborylation products (**42** and **43**) in moderate yields. In addition to pinacol esters, other borates can be conveniently obtained by trans-esterifying catechol ester intermediates. Thus, various five- and six-membered ring boronic acid derivatives were prepared in 60 to 80% yields (**44–47**), including (+)-pinacol diol (**47**) that is frequently encountered in stereoselective synthesis[54].

To our delight, when a certain amount of KF was added, α-hydroborylation of *gem*-dichloroalkanes can readily occur (see Supplementary Table 2 for details). The use of DMF-ⁿpentanol to replace DEF will further increase the reaction yields and thus was determined as the best solvent for selective hydroborylation of *gem*-dichloroalkanes. Under the modified reaction conditions, dechlorinative hydroborylation protocol can give rise to the desired products (**48–66**) in moderate to good yields. As shown in Fig. 4, phenylpropyl derivatives with both electron-withdrawing and electron-donating substituents at the *para*-position of the phenyl ring were all suitable under standard conditions. For example, a range of versatile functional groups, such as *tert*-butyl, phenyl, fluoro, and chloro on the aromatic ring uniformly gave the desired hydroborylation products (**49–52**) in 47–72% yields. Notably, both terminal alkene (**57**) and internal alkyne (**58**) can survive. Meanwhile, α-methyl substituted *gem*-dichloroalkanes also could be used for this gold-catalyzed protocol successfully, affording the corresponding products (**59–61**) in moderate yields. Different cyclic *gem*-dichloroalkanes, including

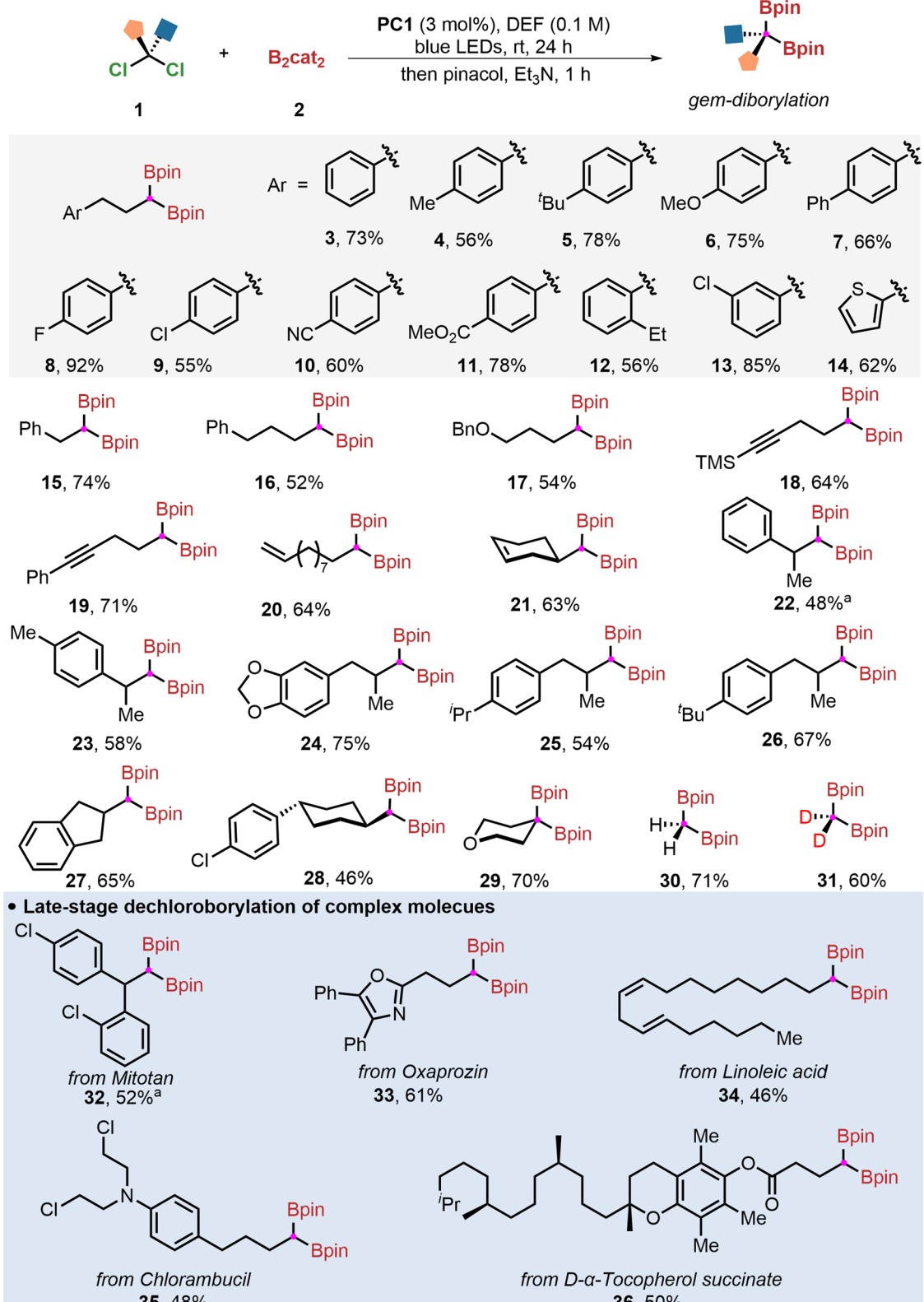

**Fig. 2 | Substrate scope of *gem*-diborylation reaction.** Reaction conditions: **PC1** (3 mol%), *gem*-dichloroalkanes **1** (0.2 mmol), $B_2cat_2$ **2** (4 equiv.), DEF (2 mL), blue LEDs, ambient temperature, 24 h; then pinacol (4 equiv.), $Et_3N$ (0.5 mL), 1 h. [a]5 equiv. $B_2cat_2$ were used.

cyclopropane, cyclopentane, and cyclohexane, can be converted into the target products (**62–64**). Complex *gem*-dichloroalkanes, such as the derivatives of mitotan (**65**) and oxaprozin (**66**), were found to be compatible, which demonstrates its generality and good functional group tolerance.

**Synthetic application**

To demonstrate synthetic utility, 10 mmol scale experiments with only 0.5 mol% catalyst loading of **PC1** were performed (Fig. 5a). The corresponding target products (**3**, **39**, and **48**) were isolated in 61% (2.27 g), 65% (1.66 g) and 48% (1.18 g), respectively. In recent years,

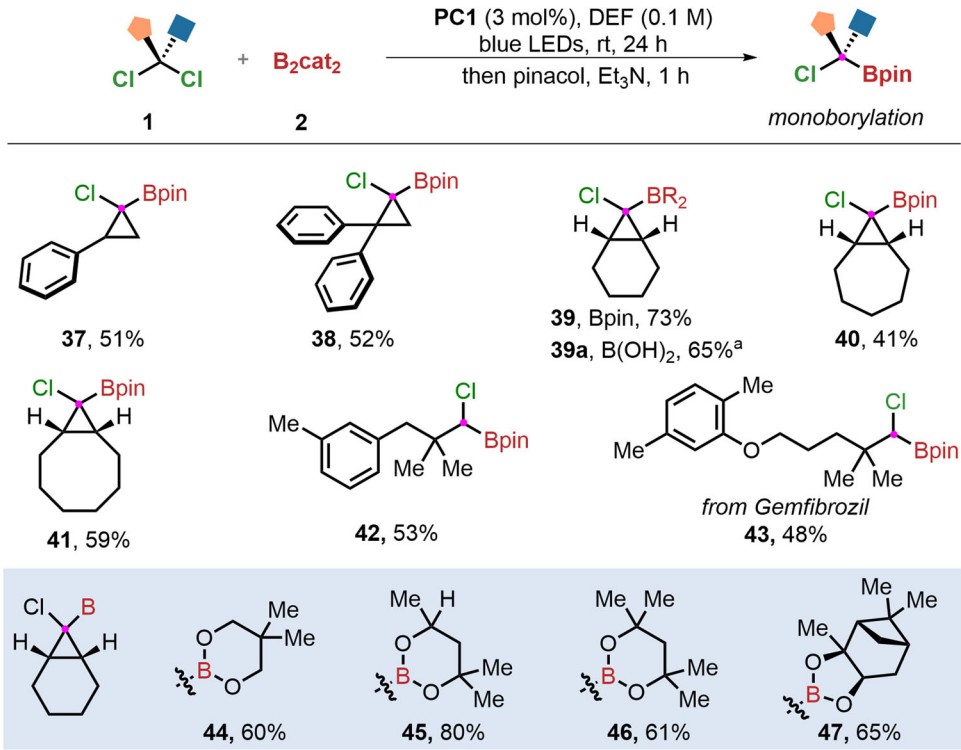

**Fig. 3 | Substrate Scope of monoborylation reaction.** Reaction conditions: **PC1** (3 mol%), *gem*-dichloroalkanes **1** (0.2 mmol), B₂cat₂ **2** (4 equiv.), DEF (2 mL), blue LEDs, ambient temperature, 24 h; then pinacol (4 equiv.), Et₃N (0.5 mL),1 h. [a]**39** (0.2 mmol), DEA (1.1 equiv.), ether, rt, ~30 min followed by 0.1 M HCl (30 min).

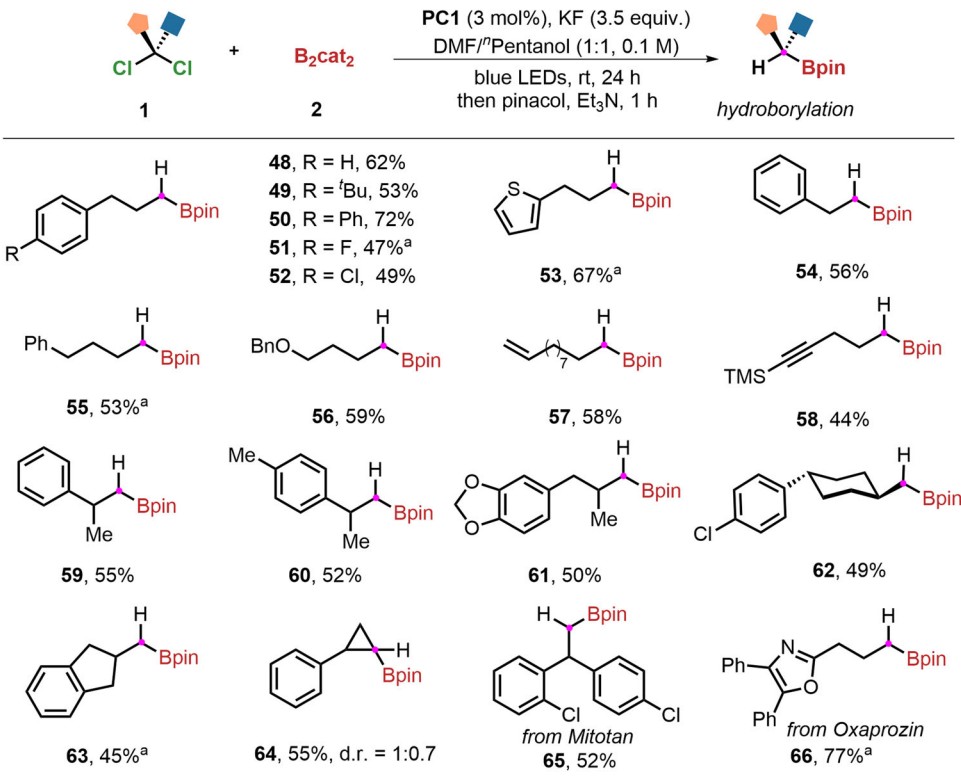

**Fig. 4 | Substrate scope of hydroborylation reaction.** Reaction conditions: **PC1** (3 mol%), *gem*-dichloroalkanes **1** (0.2 mmol), B₂cat₂ **2** (3 equiv.), KF (3.5 equiv.), DMF/ⁿPentanol (1:1, 2 mL), blue LEDs, ambient temperature, 24 h; then pinacol (4 equiv.), Et₃N (0.5 mL),1 h. [a]4 equiv. of 2-tert-butyl-1,1,3,3-tetramethyl-guanidine (BTMG) were used.

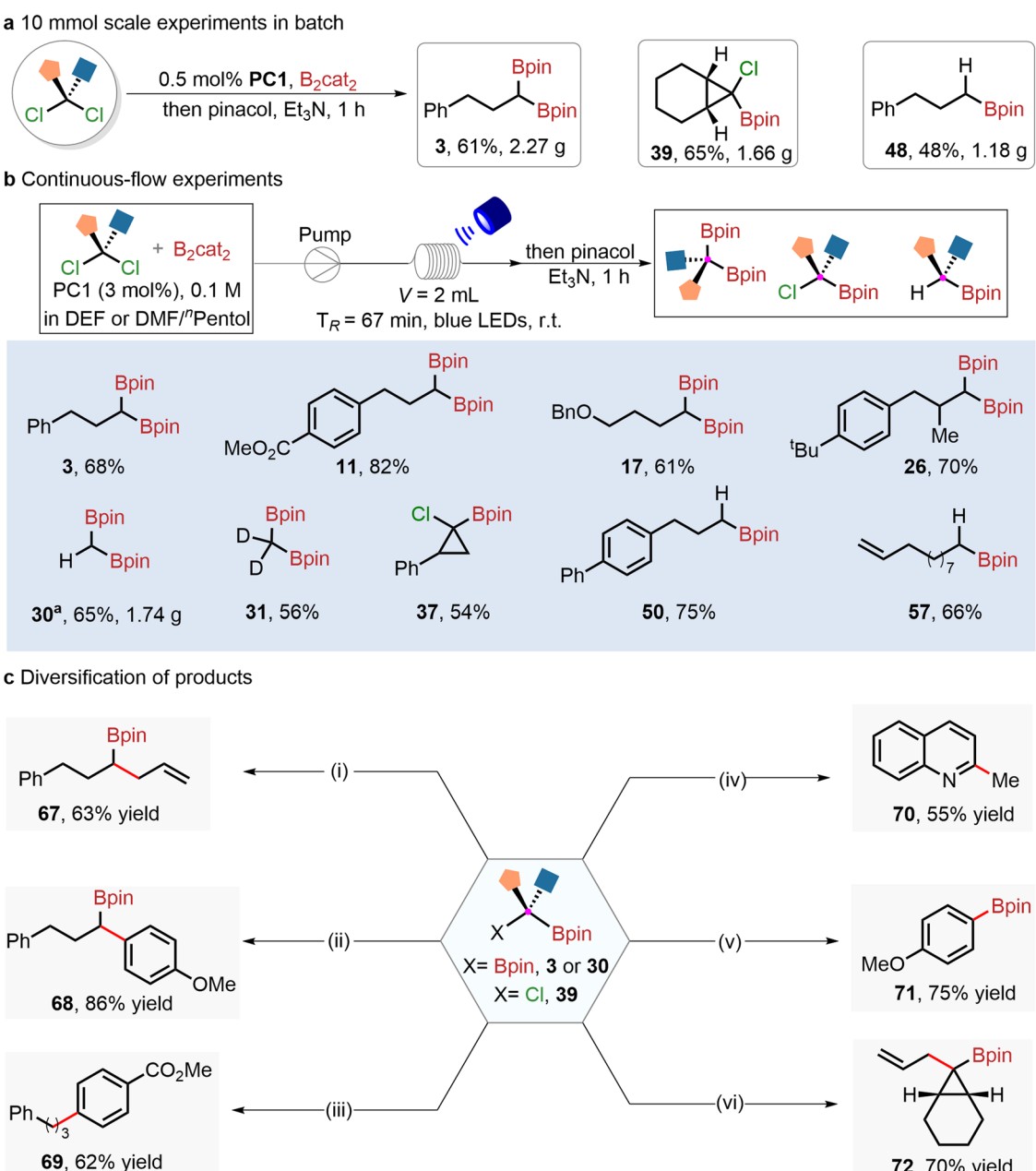

**Fig. 5 | Synthetic applications. a** Gram-scale experimental results. **b** Continuous-flow synthesis applications in dechlorinative borylation at the 1 mmol scale. [a]The reaction was performed on a 10 mmol scale with 0.5 mol% catalyst loading $T_R$, residence time; $V$, volume. **c** Diversification of products: (i) **3**, allyl chlorides, NaO$^t$Bu, THF, rt, 3 h. (ii) Pd(P$^t$Bu$_3$)$_2$ (5 mol%), **3**, 4-iodoanisole, KOH, H$_2$O/1,4-dioxane, rt, 2 h. (iii) Pd(P$^t$Bu$_3$)$_2$ (5 mol%), **3**, methyl 4-iodobenzoate, KOH, H$_2$O/1,4-dioxane, rt, 2 h. (iv) **30**, Quinoline, ZnMe$_2$, LiO$^t$Bu, toluene, 120 °C, 3 h. (v) **30**, 4-iodoanisole, NaO$^t$Bu, toluene/THF, 120 °C, 6 h. (vi) **39** allyl-MgBr, THF, −78 °C-rt, overnight.

continuous-flow microtube reactors have provided a good platform for photochemical reactions due to the improved mass and heat transfer process[55,56]. Consequently, we tried to introduce this continuous-flow synthesis technology into the gold-catalyzed divergent dechlorinative borylation of *gem*-dichloroalkanes. As shown in Fig. 5b, diborylation, monoborylation, and hydroborylation of *gem*-dichloroalkanes were successfully achieved at 1 mmol scale within a shorter reaction time. Several functionalized products (**3, 11, 17, 26, 31, 37, 50**, and **57**) were rendered in moderate to good yields. The continuous-flow technique could easily achieve 10 mmol scale production, delivering the target product (**30**) with 65% yield, which is the key building block for the synthesis of valuable drugs and natural products[57].

Importantly, these boron-containing products are versatile in terms of further synthetic transformation (Fig. 5c). For example, the borylation product (**3**) was treated with allyl chlorides in the presence of NaO$^t$Bu, the S$_N$2 substitution can readily occur to form a highly functionalized alkylboronic ester (**67**) in 63% yield. In addition, the *gem*-bis(boryl)alkane (**3**) can undergo Pd-catalyzed Suzuki-Miyaura coupling with 4-iodoanisole or methyl 4-iodobenzoate to give the coupling products (**68** and **69**) in 86% and 62% yields, respectively. Interestingly, 1,1-diborylmethanes (**30**) can be used as an alkylating reagent for the quinoline to afford C2-methylated product (**70**) in moderate yield. The reaction of 4-iodoanisole with **30** in the presence of Na$^t$OBu at 120 °C proceeded to furnish 4-methoxyphenyl boronate ester (**71**) in 75% yield. Also, treatment α-chloroboronate ester (**39**) with allyl magnesium

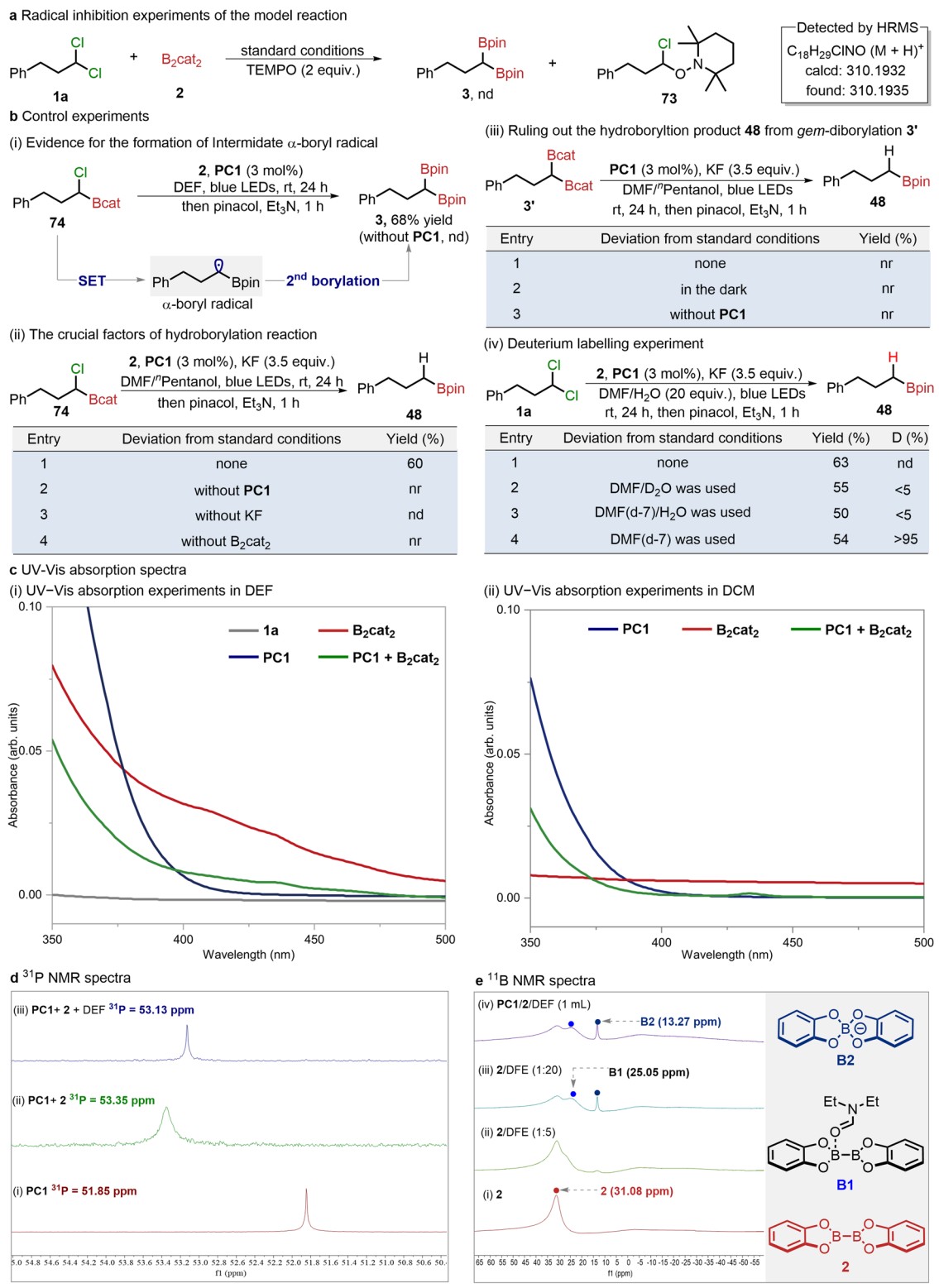

**Fig. 6 | Mechanism studies. a** Radical inhibition experiments of the model reaction. **b** Control experiments. **c** UV-Vis absorption spectra in DEF and DCM, respectively. **d** [31]P NMR monitoring experiments. **e** [11]B NMR monitoring experiments.

bromides afforded the tertiary alkylboronate esters (**72**) in 70% yield. It is an important building block for the formation of sterically hindered quaternary carbon via 1,2-metallate rearrangements[58].

## Mechanistic studies

To gain insight into the mechanism of this gold-catalyzed dechlorinative borylation process, the model reaction was performed in the presence of radical scavenger (2,2,6,6-tetramethyl-1-piperidin-1-yl)oxyl (TEMPO), no desired product (**3**) was observed and the TEMPO-trapped chloroalkyl radical adduct was successfully detected by high-resolution mass spectrometry (HRMS; Fig. 6a). To investigate the possible intermediate of dechlorinative *gem*-diborylation, we performed the following control experiments (Fig. 6b). When the α-chloroboronic esters (**74**) were subjected to the standard conditions

of diborylation, the expected *gem*-diborylated product (**3**) can be obtained in 70% isolated yield while no expected product was detected in the absence of dinuclear gold-catalyst (Fig. 6b-i). This would suggest that **PC1** is essential for the generation of α-boryl radical intermediate. Under the standard reaction conditions of hydroborylation (Fig. 6b-ii), the α-chloroboronic esters (**74**) can deliver the hydroborylated product (**48**) in 58% isolated yield, which indicates that **PC1**, $B_2cat_2$, and KF were essential for this hydroborylation process. However, *gem*-diboronic esters (**3**) cannot give rise to hydroborylation product **48** under different conditions (Fig. 6b-iii), which further demonstrated that α-chloroboronic esters would be one possible intermediate for the hydroborylation product.

Subsequently, we performed the deuterium labelling experiments to determine the hydrogen source of this hydroborylation (Fig. 6b-iv). Under the modified standard conditions of hydroborylation with DMF/$H_2O$ instead of DMF/$^n$pentanol, (3,3-dichloropropyl)benzene (**1a**) can also furnish product (**48**) in 63% yields. Surprisingly, in the presence of either DMF/$D_2O$ or DMF($d$-7)/$H_2O$, moderate yield of expected products can be detected but with little deuterium incorporation. We envisioned that once water is involved in the reaction system, the rapid H/D exchange would enable H-atom transfer prior to the D-atom transfer. It is important to find that the deuterated product (**48-D**) was isolated in 54% yield with > 95% deuterium incorporation when DMF($d$-7) was employed in the hydroborylation reaction (see Supplementary Fig. 25 for details). Moreover, the DFT calculation result indicates that the free energy barrier for H-atom transfer from water is as high as 56.9 kcal mol$^{-1}$ (see Supplementary Fig. 62). These results together suggest that the hydrogen source of hydroborylation may originate from DMF.

In addition, we measured the UV-Vis spectra of the gold catalyst and corresponding substrates in different solvents of DEF and DCM (Fig. 6c). It was found that the gold catalyst (**PC1**) has a very weak absorption in the range of 400-500 nm. However, by mixing the gold catalyst (**PC1**) with $B_2cat_2$ (**2**) in DEF for the UV−Vis measurement, a shift was observed (Fig. 6c, left). By comparison of the UV-Vis spectra in DEF and DCM, the DEF would coordinate with $B_2cat_2$ to form a new complex of DEF-ligated $B_2cat_2$[11] (Fig. 6c, right). Consequently, we speculate that a suitably bound **PC1**/DEF-ligated $B_2cat_2$ complex might be formed to enhance its absorption under irradiation by blue LEDs.

To further demonstrate the interaction between **PC1**, $B_2cat_2$, and DEF, $^{31}P$ NMR experiments were performed by using $Ph_3P$ as the external standard (Fig. 6d). When $B_2cat_2$ was added into the dinuclear gold-catalyst (**PC1**) solution in $CD_3CN$, it is found that the chemical shift can shift from 51.85 ppm to 53.35 ppm while the addition of DEF would further result in another shift from 53.35 ppm to 53.13 ppm. It would further support the formation of a bound complex between **PC1** and DEF-ligated $B_2cat_2$, consistent with previous UV-Vis spectra results. In addition, the $^{11}B$ NMR spectra of $B_2cat_2$ with different equivalents of DEF can also confirm this. As shown in Fig. 6e, the $^{11}B$ NMR signal of $B_2cat_2$ (31.08 ppm) increasingly shifted along with the addition of DEF and a relatively broad upfield signal of 25.05 ppm and sharp upfield signal of 13.27 ppm were observed when largely excess DEF (20 equiv.) was added. These new boron signals can be assigned to the ligated diboron-complex (**B1**) and Bcat$_2$ complex (**B2**)[11]. Furthermore, our DFT calculations coupled with IGMH analysis reveal that **PC1** and DEF-ligated $B_2cat_2$ can form a bound complex, characterized by a B-Cl distance of 2.05 Å and intermolecular interaction energy of -37.7 kcal mol$^{-1}$ (Fig. 7a).

Based on the above mechanistic experiments, a proposed mechanism for gold-catalyzed divergent dechloborylation is illustrated in Fig. 7a. First, the complexation of DEF and $B_2cat_2$ can generate DEF-ligated $B_2cat_2$ complex (**75**) in-situ. Its combination with photocatalyst (**PC1**) can generate a bound gold complex (**76**), which is supported by DFT, UV-Vis, and NMR. This species (**76**) then absorbs visible light to generate a triplet excited-state species (**77**).

Significantly, a single-electron transition of the excited species will not only enhance the aurophilic interaction but also provide an increasing coordination number at the Au$^I$ site[59]. The existence of a lone-pair electron on the halogen atom facilitates the dynamic coordination between *gem*-dichloroalkanes and the excited-state complex (**77**) to form an exciplex (**78**), which undergoes an inner-sphere single electron transfer (SET) process to deliver the corresponding Au(I)-Au(II) intermediate as well as chloroalkyl radical species. Owing to the stronger electrostatic interaction of chloride towards Au(I)-Au(II) than Au(I)-Au(I), the bound complex between **PC1** and DEF-ligated $B_2cat_2$ will become much weaker. Consequently, the produced chloroalkyl radical can proceed with the rapid radical attack to DEF-ligated $B_2cat_2$ to generate the α-chloroboronate species (**80**) and release Au(I)-Au(II) species (**81**) as well as the carbon-centered α-aminoalkyl radical (**82**). Subsequently, this species (**82**) is able to proceed with single electron oxidation by Au(I)-Au(II) to generate the Bcat-DEF adducts (**83**), thus completing the gold-catalyzed photoredox cycle. On the other hand, the resulting α-chloroboronate species (**80**) can similarly undergo the relay functionalization to give rise to the *gem*-diborylated products (**84**) via the second radical borylation, while hydroborylated products (**85**) via hydrogen atom transfer (HAT) with solvents. As fluoride is an important factor for a high yield in the hydroborylation, we envisioned that it would be a good mediator to interact with the boron center and DMF to generate complex **86**[60], facilitating the HAT process. To figure out the influence of fluoride anions on the HAT process, further theoretical calculations were performed (Fig. 7b). It is interesting to find that the free energy barrier of the transition state in the HAT process is 13.4 kcal mol$^{-1}$, which is much lower than that of direct HAT with DMF (**TS-2**: 17.9 kcal mol$^{-1}$) (see computational details in Supplementary Discussion). The computed energy profile shows the superiority of the HAT process assistance with the fluoride over the direct HAT process.

## Discussion

In summary, we have developed a divergent radical dechloborylation of readily available *gem*-dichloroalkanes, facilitated by dinucelar gold catalysis and visible light irradiation. A wide range of structurally diverse high-value alkyl boronic, α-chloroboronic, and *gem*-diboronic esters can be constructed from parent *gem*-dichloroalkanes in moderate to good yields (>60 examples, up to 92% yields) under mild conditions. Notable for its excellent functional group tolerance, this protocol is particularly effective for late-stage radical borylation of complex molecules and the derivatives of biologically important molecules. Thus, *gem*-diborylation, monoborylation, and hydroborylation can diversify the alkyl boronic acids, complementing the known defunctionalization radical borylations. Moreover, the utilization of continuous-flow synthesis can further improve the decholoborylation efficiency, allowing for gram-scale preparation.

## Methods
### General procedure for geminal diborylation of *gem*-dichloroalkanes
To an oven-dried 10 mL sealed tube, **PC1** (3 mol%, 0.006 mmol, 7.7 mg), *gem*-dichloroalkanes **1** (0.2 mmol), $B_2cat_2$ **2** (0.8 mmol, 190 mg, 4 equiv.) and DEF (2 mL) are successively added and the tube is back-filled with argon three times under Schlenk line. The resulting reaction mixture is vigorously stirred under the irradiation of blue LEDs (distance app. 4 cm from the bulb) at ambient temperature (the fan was used to keep the reaction temperature around ambient temperature) for 24 h. Subsequently, a solution of pinacol (0.8 mmol, 4 equiv., 94.4 mg) in triethylamine (0.5 mL) is added to the resulting crude and the reaction mixture is kept stirring at room temperature for another 1 h. Then, saturated brine water (10 mL) is added to the reaction mixture and the aqueous layer is extracted with ethyl acetate (3 × 5 mL).

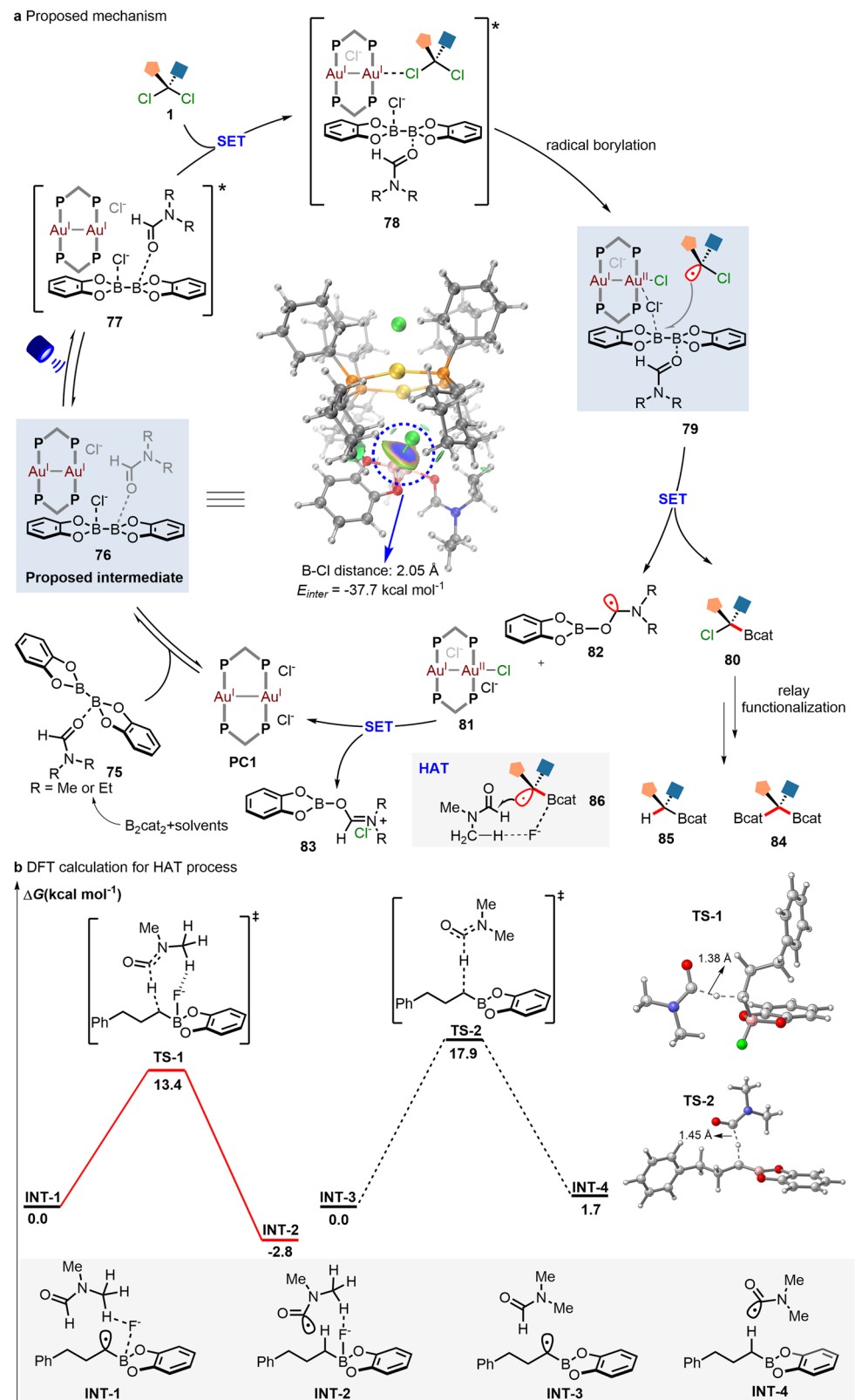

**Fig. 7 | Proposed mechanism and computational studies. a** Reaction mechanism for gold catalyzed divergent dechlorinative borylation. SET single-electron transfer, HAT hydrogen atom transfer. **b** DFT calculation results of HAT process at the PBE (D3BJ)/def2-SVP/def2-TZVP/PCM(DMF)//RI-wB97M-V/def2-TZVPP/SMD(DMF) level of theory.

The organic layers are combined, dried over Na$_2$SO$_4$, filtered, and concentrated. The crude residue is directly purified quickly by silica column chromatography (eluted with ethyl acetate/petroleum ether) to yield the desired *gem*-diborylation products.

**General procedure for monoborylation of *gem*-dichloroalkanes**
To an oven-dried 10 mL sealed tube, **PC1** (3 mol%, 0.006 mmol, 7.7 mg), *gem*-dichloroalkanes **1** (0.2 mmol), B$_2$cat$_2$ **2** (0.8 mmol, 190 mg, 4 equiv.) and DEF (2 mL) are successively added and the tube is

backfilled with argon three times under Schlenk line. The resulting reaction mixture is vigorously stirred under the irradiation of blue LEDs (distance app. 4 cm from the bulb) at ambient temperature (the fan is used to keep the reaction temperature around ambient temperature) for 24 h. Subsequently, a solution of pinacol (0.8 mmol, 4 equiv., 94.4 mg) in triethylamine (0.5 mL) is added to the resulting crude and the reaction mixture is kept stirring at room temperature for another 1 h. Then, saturated brine water (10 mL) is added to the reaction mixture and the aqueous layer is extracted with ethyl acetate (3 × 5 mL) three times. The organic layers are combined, dried over $Na_2SO_4$, filtered, and concentrated. The crude residue is directly purified quickly by silica column chromatography (eluted with ethyl acetate/petroleum ether) to yield the desired monoborylation products.

### General procedure for hydroborylation of *gem*-dichloroalkanes

To an oven-dried 10 mL sealed tube, **PC1** (3 mol%, 0.006 mmol, 7.7 mg), *gem*-dichloroalkanes **1** (0.2 mmol), $B_2cat_2$ **2** (0.6 mmol, 142 mg, 3 equiv.), KF (0.7 mmol, 41 mg, 3.5 equiv.) and DMF/$^n$Pentanol (1:1, 2 mL) are successively added and the tube is backfilled with argon three times under Schlenk line. The resulting reaction mixture is vigorously stirred under the irradiation of blue LEDs (distance app. 4 cm from the bulb) at ambient temperature (the fan is used to keep the reaction temperature around ambient temperature) for 24 h. Subsequently, a solution of pinacol (0.8 mmol, 4 equiv., 94.4 mg) in triethylamine (0.5 mL) is added to the resulting crude and the reaction mixture is kept stirring at room temperature for another 1 h. Then, saturated brine water (10 mL) is added to the reaction mixture, and the aqueous layer is extracted with ethyl acetate (3 × 5 mL) three times. The organic layers are combined, dried over $Na_2SO_4$, filtered, and concentrated. The crude residue is directly purified quickly by silica column chromatography (eluted with ethyl acetate/petroleum ether) to yield the desired hydroborylation products.

## Data availability

Data related to materials and methods, optimization of conditions, experimental procedures, mechanistic experiments, and spectra are provided in the Supplementary information. Source data containing the coordinates of the optimized structures. All data are available from the corresponding authors upon request. Source data are provided with this paper.

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

## Acknowledgements

We thank the National Natural Science Foundation of China (22122103 to J.X. and 22101130 to J.H.), the National Key Research and Development Program of China (2022YFA1503200 and 2021YFC2101901 to J.X.), Fundamental Research Funds for the Central Universities (020514380304, 020514380252 and 020514380272 to J.X.) and Postgraduate Research & Practice Innovation Program of Jiangsu Province (KYCX23_0104 to C.-L.J.) for financial support. Jie Han acknowledges the support from Xiaomi Foundation. All theoretical calculations were performed at the High-Performance Computing Center (HPCC) of Nanjing University. Jun Huang, Qingyun Fang, Duanyang Liu and Yantao Li are warmly acknowledged for their reproduction of the experimental procedures for products **3**, **21**, **39**, and **50**.

## Author contributions

C.-L.J. and J.X. conceived the work and designed the experiments. C.-L.J., H.C., and W.L. performed the experiments and analyzed the experimental data. Q.G and J.H. performed the theoretical calculations. C.-L.J. and J.X. co-wrote the manuscript with input from all the other authors.

## Competing interests

The authors declare no competing interests.
