## [Peer Review File · Nature Communications]

Dinuclear Gold-Catalyzed Divergent Dechlorinative Radical
Borylation of gem-DichloroalkanesREVIEWER COMMENTS

Reviewer #1 (Remarks to the Author):

The manuscript by Xie et al describes the borylation of dichloroalkanes in the presence of gold catalyst. The product yields are moderate and the functional group tolerability is examined. Mechanistic studies were carried out to prove the radical intermediate and the inner sphere mechanism involved in the gold complexes. DFT calculations were carried out to prove the hydrogen atom transfer reaction with the assistance of solvents. However, the lack of novelty in the work and the mechanistic studies are not convincing to meet the standard of Nature communications. Hence I regret to accept this manuscript for the publication on Nat. Communication for the points mention below.

1. The borylation of alkylchlorides is a very well established using a simple base metal catalysts such as Cu, Zn, Co, Fe, Ni and Mn. The yields are excellent in most of the reactions and there are examples for the synthesis of gem-diboroalkanes from dichloroalkanes in some of the catalytic systems listed above. A gem-diboroalkanes can be efficiently achieved by a simple metal free reaction using electro oxidations (Angew chem. Int. ed. 62, e202218179). Very recently, Liu et al. reported a very simple, scalable and efficient work for the synthesis of gem-diboroalkanes from diborodichloromethane reagents (Angew.Chem. Int. ed. 2024, 63, e202315227). In this article a sequential difunctionalization has been described with variety of functional group compatibility. Whereas, the work reported by Xie here uses Gold catalyst with 4 equivalents of B2cat2 to achieve this transformation, which is a disadvantage of this method.

Moreover, the author has published a similar work on the alkylation of gem-dichloroalkanes by gold-catalyzed using alkene as a substrate (Nat. Catalysi, 2022, 1098). I simply see this as an extension of the previous work by a small change with the substrate (B2cat2 in place of alkene) used.

2. The author claims the controllability of the reactions. I observed that the same reaction conditions were used for both monoborylation and diborylations with only change in the substrates used for both the reactions. It clearly shows that the monoborylation can be achieved only with the sterically hindered substrates, most cases has cyclopropane unit. I do not see the controllability here, rather than the diborylation reactions is difficult to achieve in case of sterically demanding substrates.

is it possible to get the monoborylated product selectively for the substrates listed in Fig.2. or the selective diborylation product for the substrates listed in Fig.3 under the standard conditions?

3. I do not see any importance of the reaction listed in Fig. 6, unless a real application on the biological systems using diboroalkanes are explored. This is to increase the number of experiments than providing any impact to the work presented. Thus it is inappropriate to emphasis on the title "biocompatibility", which will have different meaning.

4. Most of the products are isolated in moderate yields despite the use of 4 equiv of B2cat2. Why does the reaction requires huge excess of B2cat2? The higher yield is achieved only with one substrate, (compound no:8). Why does the fluorine substituted yielded high, whereas the other electron withdrawing group only provided a moderate yield, any rationalisation?

5. Mechanistic Studies:

a) Tempo experiments with B2cat2 for the radical formations does not provide any valid evidence, as the TEMPO itself reacts with B2cat2. How much of compound 73 (in %) formed in the radical reaction shown in Fig. 7 a.

b) The role of solvent in deuterium labelling experiments are not convincing, as one would expect the abstraction of proton from the water (when used as a solvent). A deuterium labelling experiments using deuterated DMF would be more appropriate.

c) Intermediate 76: The UV vis absorption studies are not convincing, as I could not see any noticeable change in the absorption spectra reported in Fig. 24 and 25 in Supporting information. Similarly the ^{31}P NMR is not informative as only a slight change in the chemical shift values are observed. and how does that suggest the possible intermediate 76?

d) If the chlorine attached to one of the boron atoms(intermediate 76), do you see any additional peak in the ^{11}B NMR?

e) How does only one peak in ^{31}P NMR is observed, when Au(I)-Au(II) are proposed to interact with alkyl radical? How does the electrophilicity of Au(II) is stronger than the Bcat in intermediate 73?

f) Do you observe any radical cross coupling products? for eg: homocoupling of alkylradical or cross coupling with radical 82?

g) KF is know to generate a sp^2 - sp^3 hybridized diboron reagent, which in turn acts as a source of B -nucleophile. Have you observed any such adducts in the hydroborylation reaction?

It is somewhat unclear that the hydrogen atom transfer occurs from α -chloroboronic esters? Why not the hydrodeborylation from the gem-diboronic esters? In general the alkylboronic esters are tend to undergo hydrodeboration reaction.

As a whole, the authors tried to decorate the work by adding numerous reactions, with the lack of novelty and efficiency, which I feel is mandatory for the publication in Nat.Communications.

Minor Corrections:

1. line 49: It should read as "Alkyl halide" not Ally halide
2. B2pin2 and B2cat2: "pin" and "cat" should be small letters
3. line 231: the yield of 68 is different from the scheme.
4. Many of the references cited did not have a complete author lists. Ref no: 12, 16, 17, 44, 53 etc.

Reviewer #2 (Remarks to the Author):

The manuscript by Xie and co-workers describes an attractive strategy based on gold-catalyzed radical dechloroborylation under blue light yielding diverse boronic ester derivatives in moderate to good yields. The methodology is based on C-Cl bond activation based on an inner sphere electron transfer mechanism that can overcome the redox potential limitation often encountered in SET photo(electro)chemistry of alkyl chlorides. However, this has been previously described by the authors for dechloroalkylation chemistry of alkenes (Nature Catalysis, 2022, 5,1098-1109). In contrast, this MS is focused on (di)borylation chemistry employing B2Cat2 as the only working boron reagent and hence needs pinacol addition to generate the targeted Bpin derivatives. It would be interesting to see a screening of different boron installers further to expand the utility and functionality of this work. Overall, MS is well described, easy to read and can be considered after the following questions can be answered.

- 1) The authors should at least comment on the specific role of DEF as a solvent besides stabilizing the boron species since DMF or DMA can also do that effectively.
- 2) The reaction is specific to B2Cat2, any reasoning for this behavior as overall 8 equiv of boron reagents are needed otherwise.
- 3) Figure 3, the scope of boronates, could this methodology be applicable to directly synthesize free boronic acid derivatives, as most of the applications are based on these species.
- 4) Figure 5: Pinacol addition part or std conditions are missing in these figures. Also check spelling of pinacol at different figure/table captions.
- 5) It is great to see that the reaction works effectively at a 10 mmole scale in batch! However, in the flow experiments, what does glass pipe mean exactly as authors have used PFA tubing (SI)? So a bit more clarity is needed and the correct terminology should be used (Conc., residence time, flow rate etc.). Moreover, a slightly bigger picture of the setup (SI) should be helpful for the readers. The residence time and overall reaction time to complete the whole batch should not be confused. Therefore, spacetime yield or throughput/h could also be mentioned. In the flow experiments (SI) there is no mention of pinacol addition to synthesize Bpin derivatives.
- 6) As the most common biological medium is water, so what about the aqueous compatibility?
- 7) Also, biocompatibility studies for monoborylation or hydroborylation reactions should also be mentioned.
- 8) Figure 7b(i) Again pinacol addition is missing. Also, to understand alpha-boryl radical contains Bpin or B2Cat in the actual mechanism will be interesting.
- 9) This work being broad and diverse, on multiple occasions, there is a deviation from standard optimized conditions whether in the scope or mechanistic studies, so I request authors to provide reasoning/comment wherever applicable.

Reviewer #3 (Remarks to the Author):

The authors reported very interesting research work. This reviewer thinks this should be regarded as a big breakthrough in radical borylation in fields of photocatalysis. The previous photocatalytic routes can address more reactive C-Br or C-I bonds, the direct use of inert C-Cl bonds is indeed challenging. The authors can successfully realize controllable dechlorinative radical borylation, one C-B bond, two C-B bonds and one C-H and one C-B

bond. A wide range of structurally diverse alkyl boronic, chloroboronic, and gem-diboronic esters can be uniformly synthesized in moderate to good yields. Furthermore, the authors have established that gold-catalyzed this work hold good biocompatibility, which can tolerate a wide range of bio-additives in the catalytic system. Based on their previous work in 2022, the authors have further improved the reaction conditions (C-C bond formatting, Nat. Catal. Ref. 37), where the HEH ester are not required any more. It enables the dinuclear gold catalysis more robust and practical. The reaction scope is very broad. A series of different kinds of gem-dichloroalkanes have been employed. With different reaction conditions, they can be used to undergo different transformations. This is an excellent work in radical borylation. The synthetic application of the resulting products further demonstrates the value of this work. A dozens useful downstream transformation can be realized. The scaled-up experiments have also been shown. Moreover, based on the reaction discovery, the authors have studied the reaction mechanism with experiments and DFT calculations. The UV-Vis, ³¹P NMR and ¹¹B NMR spectra provided evidence for the possible radical borylation process. The proposed weak interaction between B and chloride is also interesting, which is verified by the DFT calculation. Based on the synthetic advances, synthetic robustness and synthetic potential of the method, this reviewer would like to strongly recommend its publication in Nature Communication after minor revisions.

- The title should be reconsidered. The current cannot give the readers important information about gem-dichloroalkanes. This reviewer would suggest it as "Gold-Catalyzed Divergent Dechlorinative Radical Borylation of gem-Dichloroalkanes"
- Did the authors try PC1 with different counteranions?
- How about the Eosin Y as the photocatalyst?
- The manuscript has a lot of important information. This reviewer think it is not possible to cover all things in one article. This reviewer would suggest the authors give more reaction details in text for the continuous-flow synthesis applications in dechlorinative borylation.
- Gold photoredox catalysis has emerged an important area of research. The related reviews on the topic may be cited. For example: Chem. Commun., 2018, 54, 11069; Chem. Rev. 2021, 121, 14, 8868; Acc. Chem. Res. 2016, 49, 10, 2261; ChemCatChem 2023, 15, e202300974 etc

After these minor corrections, this manuscript can be accepted for publication in Nature Communications.

Point-by-point response to the reviewers

Many thanks for the three reviewers' insightful, helpful and useful suggestions and comments, which are very helpful for us to improve the quality of our manuscript. According to the reviewers' suggestions, we have tried our best to address almost all the points raised by three reviewers and we made the changes in the revised manuscript with yellow background highlight.

Reviewer #1 (Remarks to the Author):

The manuscript by Xie et al describes the borylation of dichloroalkanes in the presence of gold catalyst. The product yields are moderate and the functional group tolerability is examined. Mechanistic studies were carried out to prove the radical intermediate and the inner sphere mechanism involved in the gold complexes. DFT calculations were carried out to prove the hydrogen atom transfer reaction with the assistance of solvents. However, the lack of novelty in the work and the mechanistic studies are not convincing to meet the standard of Nature communications. Hence I regret to accept this manuscript for the publication on Nat. Communication for the points mention below.

1. The borylation of alkylchlorides is a very well established using a simple base metal catalysts such as Cu, Zn, Co, Fe, Ni and Mn. The yields are excellent in most of the reactions and there are examples for the synthesis of gem-diboroalanes from dichloroalkanes in some of the catalytic systems listed above. A gem-diborylalkanes can be efficiently achieved by a simple metal free reaction using electro oxidations (Angew chem. Int. ed. 62, e202218179). Very recently, Liu et al. reported a very simple, scalable and efficient work for the synthesis of gem-diborylalkanes from diborodichloromethane reagents (Angew. chem. Int. ed. 2024, 63, e202315227). In this article a sequential difunctionalization has been described with variety of functional group compatibility. Whereas, the work reported by Xie here uses Gold catalyst with 4 equivalents of B_2cat_2 to achieve this transformation, which is a disadvantage of this method.

Our answer: Thanks for the kind comments. Transition metal-catalyzed borylation of alkyl

chlorides in general require the use of strong organic base and most examples are dechlorinative monoborylation reactions. We found that only Prof. Marder reported an interesting case of copper-catalyzed bis(borylation) of dichloroalkanes (*ACS Catal.* **2016**, *6*, 8332–8335). In order to clearly demonstrate this point, we have put one sentence in the main text as: “For example, the Marder’s group disclosed several examples on Cu(II)-catalyzed gem-diborylation of dichloroalkanes in the presence of stoichiometric MeOK¹⁷.”

According to your comments, we have noticed the recent work for the synthesis of gem-diborylalkanes from diborodichloromethane reagents by Prof. Liu as well as Hong and quote them in the manuscript.

Ref 45: Fang, T., Wang, L., Wu, M., Qi, X. & Liu, C. Diborodichloromethane as versatile reagent for chemodivergent synthesis of *gem*-diborylalkanes. *Angew. Chem. Int. Ed.* **63**, e202315227 (2024).

Ref 46. Ning, P.-F., Wei, Y., Chen, X.-Y., Yang, Y.-F., Gao, F.-C. & Hong, K. A general method to access sterically encumbered geminal bis(boronates) via formal umpolung transformation of terminal diboron compounds. *Angew. Chem. Int. Ed.* **63**, e202315232 (2024).

Indeed, they were mainly carried out by nucleophilic substitution process. However, we reported gold-catalyzed dechlorinative borylation mainly to further explore the diversity of dinuclear gold catalysis and enrich the types of photocatalytic radical borylation reactions. Moreover, the current work is not only limited to *gem*-diborylation, but also monoborylation and hydroborylation reactions could be realized through appropriate adjustments. Notably, synthesis of the above three types of alkyl boronic esters using a single catalytic system has not been reported.

Although excess boron reagents are used, this new strategy provide a good choice to prepare alkyl boronic ester from alkyl chlorides under mild conditions with excellent functional group tolerance. Notably, Prof. Aggarwal's studies indicated that only half of the diboron reagent was consumed during the reaction (*Science* **2017**, *357*, 283-286). Additionally, Prof. Lu reported the use of 3 equivalents of boron reagent in the electrochemical catalytic dehalogenative borylation of *gem*-dibromides, yielding approximately 70% at 60°C (*Angew. Chem. Int. Ed.* **2023**, *62*, e202218179). Nickel

catalyzed decarboxylative borylation reaction reported by Baran, which also requires the use of 3.3 equivalents of diboron reagent (*Science* **2017**, 356, eaam7355).

Moreover, the author has published a similar work on the alkylation of gem-dichloroalkanes by gold-catalyzed using alkene as a substrate (*Nat. Catalysis*, 2022, 1098). I simply see this as an extension of the previous work by a small change with the substrate (B2cat2 in place of alkene) used.

Our answer: Thank you very much for your kind comments. As mentioned by the reviewer 3, “based on their previous work in 2022, the authors have further improved the reaction conditions (C-C bond formation, *Nat. Catal. Ref. 37*), where the HEH ester are not required any more. It enables the dinuclear gold catalysis more robust and practical.” In practical, the current catalytic system is simpler and more efficient rather than just replacing alkene with B2cat2. Moreover, our previous work mainly focused on the formation of C-C bonds, while the current work has achieved the more challenging construction of C-B bonds.

2. The author claims the controllability of the reactions. I observed that the same reaction conditions were used for both monoborylation and diborylations with only change in the substrates used for both the reactions. It clearly shows that the monoborylation can be achieved only with the sterically hindered substrates, most cases has cyclopropane unit. I do not see the controllability here, rather than the diborylation reactions is difficult to achieve in case of sterically demanding substrates. Is it possible to get the monoborylated product selectively for the substrates listed in Fig. 2. or the selective diborylation product for the substrates listed in Fig.3 under the standard conditions?

Our answer: Thanks for your kind suggestions. According to your suggestions, we also tried to use the substrates listed in Fig. 2 and only trace amount of monoborylated product can be detected by GC-MS. Similarly, the substrates listed in Fig.3 were employed under the standard conditions, the diborylation products can't be obtained. According to your comments, we have revised some statements about controllable borylation in the revised manuscript. In addition, one sentence was added in the revised manuscript as “We found that gold-catalyzed decarboxylative monoborylation of very sterically hindered *gem*-

dichloroalkanes can proceed smoothly, delivering the desired α -chloro alkylboronic esters in moderate to excellent yields.”

3. I do not see any importance of the reaction listed in Fig. 6, unless a real application on the biological systems using diboroalkanes are explored. This is to increase the number of experiments than providing any impact to the work presented. Thus it is inappropriate to emphasis on the title "biocompatibilitiy", which will have different meaning.

Our answer: Thanks for your kind suggestions. After careful consideration and incorporating the reviewer 2 comments on biocompatibility, we have deleted it in the manuscript and moved Fig. 6 to the Supplementary Information. Additionally, the title has been revised as “Dinuclear Gold-Catalyzed Divergent Dechlorinative Radical Borylation of gem-Dichloroalkanes”. The accompanying text has been revised as well.

4. Most of the products are isolated in moderate yields despite the use of 4 equiv of B₂cat₂. Why does the reaction requires huge excess of B₂cat₂?

Our answer: Thanks for your helpful suggestion. We conducted an equivalent screening of boron reagents and found that a minimum of three equivalents was required to achieve a moderate yield. The primary issue was incomplete conversion of *gem*-dichloroalkanes when using reduced equivalents. To address this, we increased the amount of boron reagent to enhance the yield. In addition, the role of B₂cat₂ is significant, as it can participate in reactions as a boronation reagent and interact with the solvent N,N-Diethylformamide to form B₂cat₂•DEF adducts. Notably, Prof. Aggarwal's studies indicated that only half of the diboron reagent was consumed during the reaction (*Science* **2017**, 357, 283-286). Additionally, Prof. Lu reported the use of 3 equivalents of boron reagent in the electrochemical catalytic dehalogenative borylation of *gem*-dibromides, yielding approximately 70% at 60°C (*Angew. Chem. Int. Ed.* **2023**, 62, e202218179). Nickel catalyzed decarboxylative borylation reaction reported by Baran, which also requires the use of 3.3 equivalents of diboron reagent (*Science* **2017**, 356, eaam7355).

Entry	B ₂ cat ₂ loading	Yield (%) ^b
1	1.5 equiv.	28
2	2.0 equiv.	41
3	2.5 equiv.	55
4	3.0 equiv.	62
5	3.5 equiv.	68
6	4.0 equiv.	78 (73) ^c
7	4.5 equiv.	65

^aStandard reaction conditions: **PC** (3 mol%), **1a** (0.2 mmol), B₂cat₂ **2** (x equiv.), DEF (2.0 mL), blue LEDs ($\lambda_{\text{max}}=466$ nm), ambient temperature, 24 h; then pinacol (4.0 equiv), Et₃N (0.5 mL), 1 h. ^bGC yield using biphenyl as an internal standard. ^cIsolated yield.

The higher yield is achieved only with one substrate, (compound no:8). Why does the fluorine substituted yielded high, whereas the other electron withdrawing group only provided a moderate yield, any rationalisation?

Our answer: Thanks for your helpful suggestion. As shown in the below figure, some other electron-withdrawing group (-CO₂Me or *m*-Cl) also could afford the higher yield. We speculated that it may be mainly caused by the difficulty of different products in the column chromatography separation process.

5. Mechanistic Studies:

a) Tempo experiments with B₂cat₂ for the radical formations does not provide any valid evidence, as the TEMPO itself reacts with B₂cat₂. How much of compound 73 (in %) formed in the radical reaction shown in Fig. 7 a.

Our answer: Thanks for your helpful suggestion. According to your comments, we have

tried to isolate it. However, compound **73** can only be obtained in trace yield, which detected by HR-MS.

b) The role of solvent in deuterium labelling experiments are not convincing, as one would expect the abstraction of proton from the water (when used as a solvent). A deuterium labelling experiments using deuterated DMF would be more appropriate.

Our answer: Thanks for your helpful suggestion. According to your suggestion, we performed the deuterium labelling experiments using deuterated DMF and the results were shown as following. We also added this in the manuscript as follow. One new sentence was added in the revised manuscript as “These results together suggest that the hydrogen source of hydroborylation may originate from DMF.”

(iv) Deuterium labelling experiment

Entry	Deviation from standard conditions	Yield (%)	D (%)
1	none	63	nd
2	DMF/D ₂ O was used	55	<5
3	DMF(d-7)/H ₂ O was used	50	<5
4	DMF(d-7) was used	54	>95

c) Intermediate 76: The UV vis absorption studies are not convincing, as I could not see any noticeable change in the absorption spectra reported in Fig. 24 and 25 in Supporting information.

Our answer: Thanks for your helpful suggestion. Sorry for this misunderstanding. In Supplementary Fig. 24, we measured absorption of **B₂cat₂** in DEF. In Supplementary Fig. 25, we measured absorption between *gem*-dichloroalkens **1a** and **B₂cat₂** in DEF, which indicated no difference from **B₂cat₂** absorption alone. The results indicated no interaction occurred between them. These are in the right case.

Figure 1. Absorption spectra of **2**

Figure 2. Absorption spectra of **1a** and **2**

Similarly the ^{31}P NMR is not informative as only a slight change in the chemical shift values are observed and how does that suggest the possible intermediate **76**?

Our answer: Thanks very much. All samples were detected by ^{31}P NMR spectra, using Ph_3P as the external standard. From the below figure, there are a slight change in chemical shifts among different components. On the other hand, intermediate is proposed based on the DFT calculation result. We are sorry for this mistake.

^{31}P NMR spectra of different components in CD_3CN

From the above spectra, the chemical shifts of the ^{31}P spectra are shifted to lower fields ($\delta = 53.13$ ppm), indicating the weak interaction between **PC1** and DEF-ligated **B₂cat₂**. Besides, we found that the results of UV absorption experiments were consistent with the theoretical simulation results as following. We speculated that a suitably bound **PC1**/DEF-ligated **B₂cat₂** complex (**76**) might be formed to enhance its absorption under irradiation by blue LEDs.

(a) UV-Vis absorption experiments (b) Simulated UV-vis absorption spectra

Finally, we also performed DET calculation to give explain. From the below figure, we can see the strong interaction between **PC1** and DEF-Ligated **B₂cat₂**. Specifically, there is an interaction force between the boron complex and the chlorine anion of the dinuclear gold catalyst.

Intermolecular interaction analysis between **PC1** and DEF-ligated **B₂cat₂**, and the corresponding iso-surfaces by independent gradient model based on Hirshfeld partition (IGMH)³⁰ (ISO=0.004 a.u.), where blue, green and red represent strong interaction, weak interaction, and steric effect.

According to NMR detection, UV-vs experiments and DFT calculation, we assumed that intermediate **76** was proposed. According to your important comments, we have changed the text for intermediate **76** in Fig. 7 as “Proposed intermediate”.

d) If the chlorine attached to one of the boron atoms (intermediate 76), do you see any additional peak in the ¹¹B NMR?

Our answer: Thank you very much for your helpful suggestion. The intermediate **76** is proposed based on the DFT calculation. Owing to the weak and dynamic coordination as well as a catalytic amount of Cl anion, it is difficult to directly detect it with ¹¹B NMR. In

addition, the broad peak of coordinated B_2cat_2 may be difficult to differentiate DEF or Cl anion.

^{11}B NMR spectra of the mixture of B_2cat_2 , **PC1** and DEF in $CDCl_3$

e) How does only one peak in ^{31}P NMR is observed, when Au(I)-Au(II) are proposed to interact with alkyl radical?

Our answer: Thanks for your helpful suggestion. Firstly, we mainly use ^{31}P NMR spectra to study the weak interaction between the ground state dinuclear gold catalyst and the diboron substrate instead of reaction system. The Au(I)-Au(II) intermediate is highly active and difficult to monitor using ^{31}P NMR. It was proposed based on our Nat. Cat. DFT calculations.

How does the electrophilicity of Au(II) is stronger than the Bcat in intermediate 83?

Our answer: Thanks for your helpful suggestion. Because this high intermediate hardly isolated and measured by CV, we refer some literatures to perform the DFT calculation to explain and the results as following (Glorius et al. *Chem. Eur. J.* **2018**, *24*, 17210).

$$E = -\frac{G_{298}(red) - G_{298}(ox)}{F} - E_{ref}$$

Where G_{red} and G_{ox} are Gibbs free energies for oxidized species and reduced species respectively, and F is Faraday's constant. The E_{ref} is absolute potential of saturated calomel electrode (SCE) in acetonitrile ($E_{ref} = 4.422$ V)

Table 1 Gibbs free energies and redox potential for intermediate **81** and **82**

Gibbs free energy (298K) in a.u.	Redox potential (V)
---------------------

Oxidized Species		Reduced Species		
83	-733.60	82	-733.70	-1.56
81	-4518.38	PC1	-4518.54	-0.10

From the above results, the Au(II) can be easily reduced by the intermediate **83**.

f) Do you observe any radical cross coupling products? for eg: homocoupling of alkylradical or cross coupling with radical **82**?

Our answer: Thank you very much. After the completion of gem-diborylation reaction, we conducted GC-MS monitoring and analysis on it. The results showed that no homocoupling of alkylradical or cross coupling with radical **82** was detected.

g) KF is known to generate a sp²-sp³ hybridized diboron reagent, which in turn acts as a source of B⁻ nucleophile. Have you observed any such adducts in the hydroborylation reaction?

Our answer: Thanks for your helpful suggestion. From the below spectrum, peak at $\delta = 31.98$ ppm can be attributed to DMF-ligated B₂cat₂ and the chemical shift changed, which may be influenced by **PC1**. Peak at $\delta = 14.42$ ppm can be attributed to the formation

bis(catecholato) boronate. A small amount of a new boron “ate” complex ($\delta = 8.18$ ppm) can be attributed to $B_2cat_2 \cdot KF$ adduct, which was proposed in the literature (*Angew. Chem. Int. Ed.* **2019**, *58*, 18830-18834).

^{11}B NMR spectra of the mixture of **2**, KF, **PC1** and DMF/ n Pentol in $CDCl_3$

It is somewhat unclear that the hydrogen atom transfer occurs from α -chloroboronic esters? Why not the hydrodeborylation from the gem-diboronic esters? In general the alkylboronic esters are tend to undergo hydrodeboration reaction.

Our answer: Thanks for your helpful suggestion. According to your suggestion, we synthesized this intermediate **74** according to known method (*J. Org. Chem.* **2016**, *81*, 1506-1519) and the intermediate **3'** according to known method (*Angew. Chem. Int. Ed.* **2023**, *62*, e202216356).

1. Synthesis and character of Int-74

¹H NMR (400 MHz, CDCl₃) spectra for compound Int-74

2. Synthesis and character of Int-3'

¹H NMR (500 MHz, CDCl₃) spectra for compound Int-3'

3. Control experiments

Deviation from standard conditions	Yield (%)
none	nr
in the dark	nr
without PC1	nr

We employed the intermediate **74** to standard reactions and still obtained the hydroborylated product in 60% isolated yield. Further control experiments have shown that catalysts, bases, and boron reagents are indispensable. In addition, we did not obtain the hydroboration product under the standard reaction conditions. Also, no target product was formed in the absence of light or catalyst. Accordingly, we think that it is less likely to be converted by *gem*-diboronic esters.

As a whole, the authors tried to decorate the work by adding numerous reactions, with the lack of novelty and efficiency, which I feel is mandatory for the publication in Nat.Communications.

Our answer: Thanks very much. This work has provided a new resolution to address the long-standing challenge existed in radical borylation. It not only provides a facile method for preparing various alkyl boronic esters with good functional group tolerance, but also provides some new insight for developing efficient catalytic paradigm.

Minor Corrections:

1. line 49: It should read as "Alkyl halide" not Ally halide

Our answer: Thanks for your kind suggestions, we have corrected it in the revised

manuscript.

2. B2pin2 and B2cat2: "pin" and "cat" should be small letters

Our answer: Thanks for your kind suggestions, we have corrected it in the revised manuscript.

3. line 231: the yield of 68 is different from the scheme.

Our answer: Thanks for your kind suggestions, we have rechecked the date of **68** carefully and revised in the manuscript. Sorry for this mistake.

4. Many of the references cited did not have a complete author lists. Ref no: 12, 16, 17, 44, 53 etc.

Our answer: Thanks for your kind suggestions. According to the reference format of this journal, only the first author is shown while the number of all authors is more than six.

Reviewer #2 (Remarks to the Author):

The manuscript by Xie and co-workers describes an attractive strategy based on gold-catalyzed radical dechloroborylation under blue light yielding diverse boronic ester derivatives in moderate to good yields. The methodology is based on C-Cl bond activation based on an inner sphere electron transfer mechanism that can overcome the redox potential limitation often encountered in SET photo(electro)chemistry of alkyl chlorides. However, this has been previously described by the authors for dechloroalkylation chemistry of alkenes (Nature Catalysis, 2022, 5, 1098-1109). In contrast, this MS is focused on (di)borylation chemistry employing B2Cat2 as the only working boron reagent and hence needs pinacol addition to generate the targeted Bpin derivatives. It would be interesting to see a screening of different boron installers further to expand the utility and functionality of this work. Overall, MS is well described, easy to read and can be considered after the following questions can be answered.

Our answer: Thank you very much for your positive evaluation to the synthetic values of our new synthetic methodology. To follow the reviewer's suggestion, we have unitized other boron reagent to perform gem-diborylation reaction. However, when the different boron reagents were used as a coupling partner, no desired product was obtained. We speculate that the important role of the electron-rich aromatic diol ligand, catechol, on the boron

reagent was highlighted by the failure of other boron reagents (entries 2-6). Several new sentences were added as “Switching B₂cat₂ to B₂pin₂ led to no reaction (Table 1, entry 6). We speculated that the important role of the electron-rich aromatic diol ligand, catechol, on the boron reagent favors the radical borylation.” These new results are added in the Supplementary Information.

Supplementary Table 2. Screening of boron source^a

Entry	Boron source	Yield (%) ^b
1	B ₂ cat ₂	78 (73) ^c
2	B ₂ pin ₂	0
3	B ₂ neop ₂	0
4	B ₂ hex ₂	0
5	B ₂ pai ₂	0
6	Bpin-Bdan	0

^aStandard reaction conditions: **PC1** (3 mol%), **1a** (0.2 mmol), Boron source **2** (0.8 mmol), DEF (2.0 mL), blue LEDs ($\lambda_{\text{max}}=466$ nm), ambient temperature, 24 h.

1) The authors should at least comment on the specific role of DEF as a solvent besides stabilizing the boron species since DMF or DMA can also do that effectively.

Our answer: Thanks for your helpful suggestion. Compared with DMF and DMA, DEF have the following advantages. In the *gem*-diborylation reaction, the solubility of DEF is better. After the reaction is completed, the system is separated into liquids, making extraction easier. Toxicity and water absorption are lower than DMF and DMA. From cyclic voltammetry (CV) experiments, the onset potential for DEF-ligated B₂cat₂ was lower than DMF-ligated B₂cat₂, indicating that the electronic properties of B₂cat₂ were reversed from a Lewis acid to an electron-donor complex after coordinating with the amide-based solvent

and easier participate in the radical reaction. According to your suggestions, one sentence was added as “We envisioned that the oxygen atom in the amide group DEF might coordinate with $B_2cat_2^{11}$, enabling the oxidation peak of B_2cat_2 shifted to more negative values, which can be rationalized by UV-Vis spectroscopy analysis, ^{11}B NMR detection, and cyclic voltammetry (CV) experiments (see Supplementary Fig. 34, 46, and 56). In addition, it is found that the good solubility of DEF can give better results.”

The CV data of B_2cat_2 in different solvents

2) The reaction is specific to B_2Cat_2 , any reasoning for this behavior as overall 8 equiv of boron reagents are needed otherwise.

Our answer: Thanks for your helpful suggestion. We conducted an equivalent screening of boron reagents and found that a minimum of three equivalents was required to achieve a moderate yield. The primary issue was incomplete conversion of *gem*-dichloralkanes when using reduced equivalents. To address this, we increased the amount of boron reagent to enhance the yield. In addition, the role of B_2cat_2 is significant, as it can participate in reactions as a boronation reagent and interact with the solvent N,N-Diethylformamide to form $B_2cat_2 \cdot DEF$ adducts. Notably, Prof. Aggarwal's studies indicated that only half of the diboron reagent was consumed during the reaction (*Science* **2017**, 357, 283-286). Additionally, Prof. Lu reported the use of 3 equivalents of boron reagent in the electrochemical catalytic dehalogenative borylation of *gem*-dibromides, yielding approximately 70% at 60°C (*Angew. Chem. Int. Ed.* **2023**, 62, e202218179). Nickel catalyzed decarboxylative borylation reaction reported by Baran, which also requires the use of 3.3 equivalents of diboron reagent (*Science* **2017**, 356, eaam7355).

Entry	B ₂ cat ₂ loading	Yield (%) ^b
1	1.5 equiv.	28
2	2.0 equiv.	41
3	2.5 equiv.	55
4	3.0 equiv.	62
5	3.5 equiv.	68
6	4.0 equiv.	78 (73) ^c
7	4.5 equiv.	65

^aStandard reaction conditions: **PC** (3 mol%), **1a** (0.2 mmol), B₂cat₂ **2** (x equiv.), DEF (2.0 mL), blue LEDs ($\lambda_{\text{max}}=466$ nm), ambient temperature, 24 h; then pinacol (4.0 equiv), Et₃N (0.5 mL), 1 h. ^bGC yield using biphenyl as an internal standard. ^cIsolated yield.

3) Figure 3, the scope of boronates, could this methodology be applicable to directly synthesize free boronic acid derivatives, as most of the applications are based on these species.

Our answer: Thanks for your helpful suggestion. According to your comments, we conducted the following experiment and updated this results in the manuscript. Several new sentences were added in the revised manuscript as “The Bpin ester can be conveniently hydrolyzed to the corresponding boronic acid (**39a**). This allows the conversion of carboxylic acids to their boron-bioisomers to identify compounds with excellent pharmacodynamic or kinetic properties.”

7-Chlorobicyclo[4.1.0]heptan-7-ylboronic acid (39a)

¹H NMR (400 MHz, CDCl₃) δ 4.47 (s, 1H), 2.03 – 1.87 (m, 2H), 1.78 – 1.56 (m, 2H), 1.56 – 1.41 (m, 2H), 1.42 – 1.31 (m, 2H), 1.31 – 1.20 (m, 2H). ¹³C NMR (126 MHz, CDCl₃) δ 21.4, 21.0, 18.8. ¹¹B NMR (160 MHz, CDCl₃) δ 31.1. HRMS m/z (ESI) calcd for C₇H₁₁BClO₂ (M – H)⁺: 173.0546; found: 173.0538.

¹H NMR (400 MHz, CDCl₃) spectra for compound **39a**

¹³C NMR (126 MHz, CDCl₃) spectra for compound **39a**

^{11}B NMR (160 MHz, CDCl_3) spectra for compound **39a**

4) Figure 5: Pinacol addition part or std conditions are missing in these figures. Also check spelling of pinacol at different figure/table captions.

Our answer: Thanks for your helpful suggestion. According to your comments, we have corrected them in the revised manuscript. B_2pin_2 instead of B_2Pin_2 , B_2cat_2 instead of B_2Cat_2 .

5) It is great to see that the reaction works effectively at a 10 mmole scale in batch! However, in the flow experiments, what does glass pipe mean exactly as authors have used PFA tubing (SI)? So a bit more clarity is needed and the correct terminology should be used (Conc., residence time, flow rate etc.). Moreover, a slightly bigger picture of the setup (SI) should be helpful for the readers. The residence time and overall reaction time to complete the whole batch should not be confused. Therefore, spacetime yield or throughput/h could also be mentioned. In the flow experiments (SI) there is no mention of pinacol addition to synthesize Bpin derivatives.

Our answer: Thanks for your positive evaluation and helpful suggestion. According to your comments, we have carefully revised and standardized the descriptions related to continuous flow in both the manuscript and supplementary information. We appreciate your guidance and feedback, which has helped us improve the clarity and professionalism of our work.

Fig. 5 | Synthetic applications. a, Gram-scale experimental results. b, Continuous-flow synthesis applications in dechlorinative borylation.

Supplementary Figure 3. Continuous-flow setup

General Procedure E: gold-catalyzed *gem*-diborylation, monoborylation and hydroborylation of *gem*-dichloroalkanes in continuous-flow

In the glovebox, the corresponding *gem*-dichloroalkanes **1** (1.0 mmol), **2** (4.0 equiv., 0.95 g), **PC1** (3 mol%, 38.5 mg), DEF (10 mL) are mixed in a 25 mL flask and the flask is swirled to achieve homogeneity. The liquid is then taken up with a syringe and mounted on a syringe pump. As shown in Supplementary Figure 1, the flow apparatus is purged with degassed argon to remove the air first. The syringe is connected to the flow apparatus with a back-pressure regulator. The tubing (HPFA, O.D. 1/16", I.D. 0.03", 5.2 m, volume = 2.0 mL) is rounded on a glass cylinder (I.D. = 5.0 cm). The reaction is placed into the center of 45 W blue LED at ambient temperature (the temperature was controlled by fan and the distance between the tubing and light is around 4 cm). The flow apparatus itself is set up with $T_R = 67$ min, flow rate = 0.03 mL/min. After approximately 15 min of equilibration, a solution of pinacol (4 mmol, 4.0 equiv., 0.472 g) in triethylamine (2.5 mL) is added to the resulting crude and the reaction mixture is kept stirring at room temperature for another 1 h. Finally, the solutions are diluted with H₂O and ethyl acetate. The layers are separated and the aqueous layer is extracted with ethyl acetate three times. The combined organic layers are washed with brine, dried with Na₂SO₄, filtered, and evaporated. The crude is purified via column chromatography on silica gel to afford the corresponding *gem*-

diborylation or monoborylation product.

In the glovebox, the corresponding *gem*-dichloroalkanes **1** (1.0 mmol), **2** (3.0 equiv., 0.71 g), **PC1** (3 mol%, 38.5 mg) and BTMG (4.0 equiv., 0.55 mL), DMF/*n*Pentanol (1:1, 10 mL) are mixed in a 25 mL flask and the flask is swirled to achieve homogeneity. The liquid is taken up with a syringe and mounted on a syringe pump. As shown in Supplementary Figure 1, the flow apparatus is purged with degassed argon to remove the air first. The syringe is connected to the flow apparatus with a back-pressure regulator. The tubing (HPFA, O.D. 1/16", I.D. 0.03", 5.2 m, volume = 2.0 mL) is rounded on a glass cylinder (I.D. = 5.0 cm). The reaction is placed into the center of 45 W blue LED at ambient temperature (the temperature was controlled by fan and the distance between the tubing and light is around 4 cm). The flow apparatus itself is set up with $T_R = 67 \text{ min}$, flow rate = 0.03 mL/min. After approximately 15 min of equilibration, a solution of pinacol (4 mmol, 4.0 equiv., 0.472 g) in triethylamine (2.5 mL) is added to the resulting crude and the reaction mixture is kept stirring at room temperature for another 1 h. Finally, the solutions are diluted with H_2O and ethyl acetate. The layers are separated and the aqueous layer is extracted with ethyl acetate three times. The combined organic layers are washed with brine, dried with Na_2SO_4 , filtered, and evaporated. The crude is purified via column chromatography on silica gel to afford the corresponding hydroborylation product.

6) As the most common biological medium is water, so what about the aqueous compatibility?

Our answer: Thanks for your helpful suggestion. According your comments, we added water to the reaction system and the reaction result is as follows. These new results were added to the manuscript.

7) Also, biocompatibility studies for monoborylation or hydroborylation reactions should also mentioned.

Our answer: Thanks for your helpful suggestion. According your comments, we examined the biocompatibility of hydroboration and monoboration, respectively. These new results were added to the manuscript.

Investigation of biocompatibility. Standard reaction conditions. After the reaction was complete, the reaction mixture was analyzed and the isolated yield was shown above.
^aRecovery yields of functionalized additives were given in parentheses.

8) Figure 7b(i) Again pinacol addition is missing. Also, to understand alpha-boryl radical contains Bpin or Bcat in the actual mechanism will be interesting.

Our answer: Thanks for your helpful suggestion. We have revised the Fig 7b(i) in the manuscript. Based on our understanding, the alpha-chloroboronic esters is relatively stable, so we initially used it as an intermediate for mechanism verification. In fact, the key intermediate formed during the *gem*-diboration and hydroboration reactions is α -ClBcat intermediate. So we synthesized this intermediate **74** according to known method (*J. Org. Chem.* **2016**, *81*, 1506-1519) and conducted the following mechanism experiments. 1. Synthesis and character of Int-74

^1H NMR (400 MHz, CDCl_3) spectra for compound **Int-74**

2. Control experiments

As shown in above tables, we employed the intermediate **74** to standard reactions and obtained the *gem*-diborylated product **3** and hydroborylated product **48** with 68 and 60% isolated yield, respectively. These results further indicate that $\alpha\text{-ClBcat}$ is the key intermediate in the gold-catalyzed dechloroboration.

9) This work being broad and diverse, on multiple occasions, there is a deviation from standard optimized conditions whether in the scope or mechanistic studies, so I request

authors to provide reasoning/comment wherever applicable.

Our answer: Thanks for your helpful suggestion. According to your comments, we have added some footnotes in the tables and figures when there is a deviation from standard optimized conditions. For the hydroborylation reaction, one sentence was used as “The use of DMF-npentanol to replace DEF will further increase the reaction yields and thus was determined as the best solvent for selective hydroborylation of gem-dichloroalkanes.”

Reviewer #3 (Remarks to the Author):

The authors reported very interesting research work. This reviewer thinks this should be regarded as a big breakthrough in radical borylation in fields of photocatalysis. The previous photocatalytic routes can address more reactive C-Br or C-I bonds, the direct use of inert C-Cl bonds is indeed challenging. The authors can successfully realize controllable dechlorinative radical borylation, one C-B bond, two C-B bonds and one C-H and one C-B bond. A wide range of structurally diverse alkyl boronic, chloroboronic, and gem-diboronic esters can be uniformly synthesized in moderate to good yields. Furthermore, the authors have established that gold-catalyzed this work hold good biocompatibility, which can tolerate a wide range of bio-additives in the catalytic system. Based on their previous work in 2022, the authors have further improved the reaction conditions (C-C bond formatting, Nat. Catal. Ref. 37), where the HEH ester are not required any more. It enables the dinuclear gold catalysis more robust and practical. The reaction scope is very broad. A series of different kinds of gem-dichloroalkanes have been employed. With different reaction conditions, they can be used to undergo different transformations. **This is an excellent work in radical borylation.** The synthetic application of the resulting products further demonstrates the value of this work. A dozens useful downstream transformation can be realized. The scaled-up experiments have also been shown. Moreover, based on the reaction discovery, the authors have studied the reaction mechanism with experiments and DFT calculations. The UV-Vis, ³¹P NMR and ¹¹B NMR spectra provided evidence for the possible radical borylation process. The proposed weak interaction between B and chloride is also interesting, which is verified by the DFT calculation. **Based on the synthetic advances, synthetic robustness and synthetic potential of the method, this reviewer would**

like to strongly recommend its publication in Nature Communication after minor revisions.

Our answer: Thank you very much for your insightful comments and positive recommendation.

1. The title should be reconsidered. The current cannot give the readers important information about gem-dichloroalkanes. This reviewer would suggest it as “Gold-Catalyzed Divergent Dechlorinative Radical Borylation of gem-Dichloroalkanes”

Our answer: Thanks a lot for the reviewer’s meticulous inspection. The title has been revised as “Dinuclear Gold-Catalyzed Divergent Dechlorinative Radical Borylation of gem-Dichloroalkanes”. The accompanying text has been revised as well.

2. Did the authors try PC1 with different counteranions?

Our answer: Thanks for your helpful suggestion. According to your comments, we try PC1 with different counteranions to test the influence of *gem*-diborylation and the results are shown below. We also added this table in the Supplementary Information.

Entry	Photocatalyst	Yield (%) ^b
1	[Au(dcpm)Br] ₂	40
2	[Au(dcpm)PF ₆] ₂	57
3	[Au(dcpm)BF ₄] ₂	53
4	[Au(dcpm)SCN] ₂	25

^aReaction conditions: **1a** (0.2 mmol), **2** (0.8 mmol), **PC** (3 mol%), DEF (2.0 mL), blue LEDs ($\lambda_{\text{max}}=466$ nm), ambient temperature, 24 h; then pinacol (4.0 equiv), Et₃N (0.5 mL), 1 h.

^bIsolated yield.

3. How about the Eosin Y as the photocatalyst?

Our answer: Thanks for your helpful suggestion. According to your comments, we utilized Eosin Y to perform the *gem*-diborylation reaction, unfortunately, no desired products **3** was obtained. This result was added in the entry 6 in Table 1.

4. The manuscript has a lot of important information. This reviewer think it is not possible

to cover all things in one article. This reviewer would suggest the authors give more reaction details in text for the continuous-flow synthesis applications in dechlorinative borylation.

Our answer: Thanks for your helpful suggestion. According to your comments, we have carefully revised and standardized the descriptions related to continuous flow in both the manuscript and supplementary information. We appreciate your guidance and feedback, which has helped us improve the clarity and professionalism of our work.

Fig. 5 | Synthetic applications. a, Gram-scale experimental results. b, Continuous-flow synthesis applications in dechlorinative borylation.

Supplementary Figure 4. Continuous-flow setup

General Procedure E: gold-catalyzed *gem*-diborylation, monoborylation and hydroborylation of *gem*-dichloroalkanes in continuous-flow

In the glovebox, the corresponding *gem*-dichloroalkanes **1** (1.0 mmol), **2** (4.0 equiv., 0.95 g), **PC1** (3 mol%, 38.5 mg), DEF (10 mL) are mixed in a 25 mL flask and the flask is swirled to achieve homogeneity. The liquid is then taken up with a syringe (50 mL) and mounted on a syringe pump. As shown in Supplementary Figure 1, the flow apparatus is purged with

degassed argon to remove the air first. The syringe is connected to the flow apparatus with a back-pressure regulator. The tubing (HPFA, O.D. 1/16", I.D. 0.03", 5.2 m, volume = 2.0 mL) is rounded on a glass cylinder (I.D. = 5.0 cm). The reaction is placed into the center of 45 W blue LED at ambient temperature (the temperature was controlled by fan and the distance between the tubing and light is around 4 cm). The flow apparatus itself is set up with $T_R = 67$ min, flow rate = 0.03 mL/min. After approximately 15 min of equilibration, a solution of pinacol (4 mmol, 4.0 equiv., 0.472 g) in triethylamine (2.5 mL) is added to the resulting crude and the reaction mixture is kept stirring at room temperature for another 1 h. Finally, the solutions are diluted with H₂O and ethyl acetate. The layers are separated and the aqueous layer is extracted with ethyl acetate three times. The combined organic layers are washed with brine, dried with Na₂SO₄, filtered, and evaporated. The crude is purified via column chromatography on silica gel to afford the corresponding *gem*-diborylation or monoborylation product.

In the glovebox, the corresponding *gem*-dichloroalkanes **1** (1.0 mmol), **2** (3.0 equiv., 0.71 g), **PC1** (3 mol%, 38.5 mg) and BTMG (4.0 equiv., 0.55 mL), DMF/*n*Pentanol (1:1, 10 mL) are mixed in a 25 mL flask and the flask is swirled to achieve homogeneity. The liquid is taken up with a syringe (50 mL) and mounted on a syringe pump. As shown in Supplementary Figure 1, the flow apparatus is purged with degassed argon to remove the air first. The syringe is connected to the flow apparatus with a back-pressure regulator. The tubing (HPFA, O.D. 1/16", I.D. 0.03", 5.2 m, volume = 2.0 mL) is rounded on a glass cylinder (I.D. = 5.0 cm). The reaction is placed into the center of 45 W blue LED at ambient temperature (the temperature was controlled by fan and the distance between the tubing and light is around 4 cm). The flow apparatus itself is set up with $T_R = 67$ min, flow rate = 0.03 mL/min. After approximately 15 min of equilibration, a solution of pinacol (4 mmol, 4.0 equiv., 0.472 g) in triethylamine (2.5 mL) is added to the resulting crude and the reaction

mixture is kept stirring at room temperature for another 1 h. Finally, the solutions are diluted with H₂O and ethyl acetate. The layers are separated and the aqueous layer is extracted with ethyl acetate three times. The combined organic layers are washed with brine, dried with Na₂SO₄, filtered, and evaporated. The crude is purified via column chromatography on silica gel to afford the corresponding hydroborylation product.

5. Gold photoredox catalysis has emerged an important area of research. The related reviews on the topic may be cited. For example: *Chem. Commun.*, 2018, 54, 11069; *Chem. Rev.* 2021, 121, 14, 8868; *Acc. Chem. Res.* 2016, 49, 10, 2261; *ChemCatChem* 2023, 15, e202300974 etc

Our answer: Thank you very much. According to your suggestion, we have added the references in the manuscript as following:

37 Akram, M. O., Banerjee, S., Saswade, S. S., Bedi, V. & Patil, N. T. Oxidant-free oxidative gold catalysis: the new paradigm in cross-coupling reactions. *Chem. Commun.* **54**, 11069-11083 (2018).

38 Hopkinson, M. N., Tlahuext-Aca, A. & Glorius, F. Merging Visible light photoredox and gold catalysis. *Acc. Chem. Res.* **49**, 2261-2272 (2016).

39 Witzel, S., Hashmi, A. S. K. & Xie, J. Light in gold catalysis. *Chem. Rev.* **121**, 8868-8925 (2021).

59 Han, J. & Xie, J. Inner-sphere single electron transfer in polynuclear gold photocatalysis. *ChemCatChem* **15**, e202300974 (2023).

After these minor corrections, this manuscript can be accepted for publication in Nature

Communications.

Our answer: Thank you very much for your insightful comments and positive recommendation.

REVIEWERS' COMMENTS

Reviewer #1 (Remarks to the Author):

The authors addressed all the comments raised by me and other two reviewers in very convincing way. They have done a lot more of additional experiments to improve the manuscript suitability. Hence, in this form I am pleased to accept this manuscript for publication in Nat. Commun.

Reviewer #2 (Remarks to the Author):

I am satisfied with the replies and modifications in the revised version of this MS.

Point-by-point response to the reviewers

Reviewer #1 (Remarks to the Author):

The authors addressed all the comments raised by me and other two reviewers in very convincing way. They have done a lot more of additional experiments to improve the manuscript suitability. Hence, in this form I am pleased to accept this manuscript for publication in Nat. Commun.

Our answer: Thank you very much for your positive comments.

Reviewer #2 (Remarks to the Author):

I am satisfied with the replies and modifications in the revised version of this MS.

Our answer: Thank you very much for your positive comments.